# Scaling Laws of Synthetic Data for Language Models

Zeyu Qin[12], Qingxiu Dong[13], Xingxing Zhang[1], Li Dong[1], Xiaolong Huang[1],

Ziyi Yang[1], Mahmoud Khademi[1], Dongdong Zhang[1], Hany Hassan Awadalla[1],

Yi R. Fung[2], Weizhu Chen[1], Minhao Cheng[4], Furu Wei[1]

[1]Microsoft, [2]HKUST, [3]Peking University, [4]Penn State University

## Abstract

Large language models (LLMs) achieve strong performance across diverse tasks, driven by high-quality web data used in pre-training. However, recent studies indicate web data is rapidly depleting. Synthetic data emerges as a promising alternative, but it remains unclear whether synthetic datasets exhibit predictable scalability comparable to raw pre-training data. In this work, we systematically investigate scaling laws of synthetic data by introducing SYNTHLLM, a scalable framework that transforms pre-training corpora into diverse, high-quality synthetic datasets. Our approach achieves this by automatically extracting and recombining high-level concepts across multiple documents using a graph algorithm. Key findings from our experiments with SYNTHLLM on math domain include: (1) SYNTHLLM generates synthetic data that reliably adheres to *rectified scaling law* across various model sizes; (2) Performance gains gradually diminish near 300B tokens; and (3) Larger models approach optimal performance with fewer training tokens. For instance, an 8B model peaks at 1T tokens, while a 3B model requires 4T. Moreover, comparisons with existing synthetic data generation and augmentation methods demonstrate that SYNTHLLM achieves superior performance and scalability. Our findings highlight synthetic data as a scalable and reliable alternative to raw pre-training data, offering a viable path toward continued improvement in model performance.

## 1 Introduction

Large language models (LLMs) achieve remarkable capabilities, primarily driven by vast amounts of high-quality pre-training data, which provide the foundational knowledge and generalization capacity necessary for their broad applicability (Radford et al., 2019; Brown et al., 2020). This relationship is further reinforced by scaling laws (Kaplan et al., 2020; Hoffmann et al., 2022; Muennighoff et al., 2023), indicating that increasing both model size and data volume typically yields predictable and consistent performance improvements.

However, recent studies (Villalobos et al., 2024; Muennighoff et al., 2023) indicate that the supply of high-quality pre-training data is rapidly diminishing. As web-scraped textual corpora approach saturation, conventional scaling methods may face diminishing returns, potentially slowing future progress. Therefore, it becomes crucial to explore more principled approaches that better harness the potential of existing corpora. Synthetic data (Gunasekar et al., 2023; Li et al., 2023; Liu et al., 2024; Maini et al., 2024; Long et al., 2024; Abdin et al., 2024) emerges as a promising alternative, providing a means to amplify and extend the utility of human-generated data. Yet, it remains unclear whether synthetic data follows predictable scaling laws akin to those established for natural pre-training data (Kaplan et al., 2020; Hoffmann et al., 2022). This raises a critical question: *Can synthetic data sustain scalable performance improvements, or are there fundamental limitations?* Understanding this is essential to assessing the viability of synthetic data as a long-term solution to data scarcity.

To investigate the scaling laws of synthetic data, we aim to develop a scalable approach for generating synthetic data at the scale. Conventional synthetic dataset generation methods rely heavily on limited human-annotated seed examples from target domains (Wang et al., 2022; Tang et al.; Yu et al., 2024; Mitra et al., 2023; Toshniwal et al., 2025; Ge et al., 2024; Li et al., 2024a), fundamentally restricting the diversity and scalability of the resulting datasets. In contrast, pre-training corpora—vast and inherently diverse—remain an underutilized resource for synthetic data generation. To address this gap, we introduce SYNTHLLM, a scalable framework designed to systematically transform pre-training data into high-quality synthetic datasets.

SYNTHLLM operates through three distinct stages. First, it autonomously identifies and filters high-quality web documents from the target domain (e.g., mathematics). Second, using these selected reference documents, it generates diverse, large-scale questions or prompts through open-source LLMs employing three complementary methods designed to progressively enhance diversity. Finally, the method produces corresponding answers to these questions, again utilizing open-source LLMs. Previous approaches typically used direct question extraction (Yue et al., 2024; Zhou et al., 2024; Yuan et al., 2025) or document back-translation (Li et al., 2024). However, these methods have inherent scalability limitations, constrained by either the finite number of reference documents containing quality questions or the necessity of dedicated back-translation model training. SYNTHLLM goes beyond *direct extraction* by automatically extracting and randomly combining high-level concepts from multiple documents with a graph algorithm, while establishing grounding across diverse reference documents. Detailed ablation studies highlight the superior performance and scalability gained by our methods.

We apply SYNTHLLM to math reasoning domain to study the scaling laws of synthetic data. We observe the following: **(1)** Synthetic data consistently follow rectified scaling law (Lin et al., 2024) across various model sizes, as illustrated in Figure 1. **(2)** Performance gains start to diminish once the amount of synthetic data exceeds approximately 300B tokens. **(3)** Larger models reach optimal performance more quickly compared to smaller ones. For instance, the 8B model requires only 1T tokens to achieve its best performance, whereas the 3B model needs 4T tokens. These results underscore the critical importance of scaling synthetic data. Even with limited pre-training data, systematically expanding the synthetic dataset yields sustained and predictable performance gains, reinforcing its role as a scalable solution to reduce reliance on human-generated data while enhancing model capabilities.

## 2 Scaling Law of Synthetic Data

In this section, we first investigate and demonstrate the scaling properties of synthetic data. Details of the approach employed for synthetic data generation (i.e., SYNTHLLM) will be presented in Section 3.

### 2.1 Scaling Laws for Organic Data

The scaling law for pre-training on organic data has been extensively studied in the literature (Kaplan et al., 2020; Hoffmann et al., 2022). In particular, the predictive loss is a parametric function of the model parameter size $N$ and data size $D$ (i.e., number of tokens) following the power-law form (Hoffmann et al., 2022):

$$L(N, D) = \frac{A}{N^\alpha} + \frac{B}{D^\beta} + \hat{E} \tag{1}$$

The function above models the scaling loss $L(N, D)$—typically measured on a validation set—as a relationship between the number of model parameters $N$ and training tokens $D$, based on classical risk decomposition. This decomposition splits the final loss into multiple terms, each corresponding to a distinct source of error. Here, $N$ denotes the LLM parameter count, and $D$ represents the number of training tokens. The irreducible term $E$ reflects the minimal achievable loss inherent to the ideal generative process over natural text distributions. The decay exponents $\alpha$ and $\beta$ control how rapidly the loss diminishes as $N$ and $D$ increase, respectively. The coefficients $A$ and $B$, typically influenced by model

architecture and data characteristics, regulate the rate of loss convergence. All parameters $\alpha, \beta, A, B$, and $E$ are learned by fitting the scaling curve to empirical data.

Hernandez et al. (2021) explore scaling laws for transfer learning, specifically the transition from unsupervised pre-training to fine-tuning. Their results indicate that fine-tuning pre-trained models is more data-efficient than training from scratch, highlighting the importance of pre-training in shaping scaling behavior. Building upon this, Lin et al. (2024) further analyze scaling laws in fine-tuning scenarios and introduce the *Rectified Scaling Law*:

$$L(D) = \frac{B}{D_l + D^\beta} + E \tag{2}$$

In the following, we illustrate how the rectified scaling law is obtained from the *vanilla* scaling law in Eq. (1). Compared to the original formulation in Eq. (1), we first remove the term associated with model parameters, as the LLM remains fixed during fine-tuning. Consequently, we derive the following marginalized form of Eq. (1):

$$L(D) = \frac{B}{D^\beta} + E \tag{3}$$

Here, $L(D)$ denotes the validation metric for evaluating the LLM's performance on downstream tasks. The irreducible term $E$ represents the optimal performance achievable by the model as data size approaches infinity, reflecting both inherent data-distribution loss and model capacity constraints. Compared to Eq. (3), the Rectified Scaling Law introduces an additional parameter, $D_l$, termed the *pre-learned data size*. This quantifies the latent knowledge relevant to the downstream task acquired during pre-training. Including $D_l$ captures the empirical observation that fine-tuning pretrained models yields better performance than training from scratch. Consequently, we obtain the final formulation in Eq. (2). Specifically, the term $\frac{B}{D_l}$ characterizes the initial performance advantage provided by pre-training. Lin et al. (2024) demonstrate that this refinement provides a more accurate depiction of how pre-training influences downstream task performance during fine-tuning.

## 2.2 Scaling Laws for Synthetic Data

Prior studies (Kaplan et al., 2020; Hoffmann et al., 2022; Hernandez et al., 2021; Lin et al., 2024) have primarily investigated scaling laws of organic data. To our knowledge, this is the first systematic exploration and empirical validation of scaling laws for synthetic data. We employ SYNTHLLM (Section 3) to generate data from a carefully filtered subset of the pre-training corpus, ensuring both scale and diversity (see Section 4.3 for details). Using the full synthetic dataset from the three methods in SYNTHLLM, we conduct continued training on LLMs of varying sizes (Llama-3.2-1B, Llama-3.2-3B, and Llama-3.1-8B) with progressively larger subsets (100K, 200K, 400K, 800K, 1.6M, 3.2M, and 7.4M examples). In this work, we main focus on the mathematical reasoning domain due to its well-established evaluation protocols and metrics. Additionally, we conduct preliminary experiments in the *coding domain*, and present detailed results in Section 4.4. Model performance is assessed by reporting error rates on the MATH dataset (Hendrycks et al., 2021), which covers a wide range of mathematical subjects and difficulty levels (1–5). We use error rate as the evaluation metric, where lower error rates indicate better performance. Further details on datasets and experiments are provided in Section 4.1.

**Overall Trend.** As shown in Figure 1, *synthetic data generated by* SYNTHLLM *adheres to the rectified scaling law* (Equation 2). The $x$-axis represents the synthetic dataset size, measured in number of tokens, while the $y$-axis depicts the error rate on the MATH dataset. Both axes employ logarithmic scaling, consistent with previous work (Kaplan et al., 2020). The green points denote the data sizes utilized to fit the scaling laws, whereas the red points serve to evaluate the predictive accuracy of the fitted curves. We additionally attempted fitting our data using Equation 3, but found that it fails to accurately model our results (see Section C.2). We hypothesize this discrepancy arises because the base models employed in our experiments already possess certain mathematical problem-solving capabilities acquired during pre-training.

**Model Size Matters** The decay exponent $\beta$ in Eq. (2) quantifies how quickly the error rate decreases as dataset size grows; a larger $\beta$ indicates a more rapid improvement given same

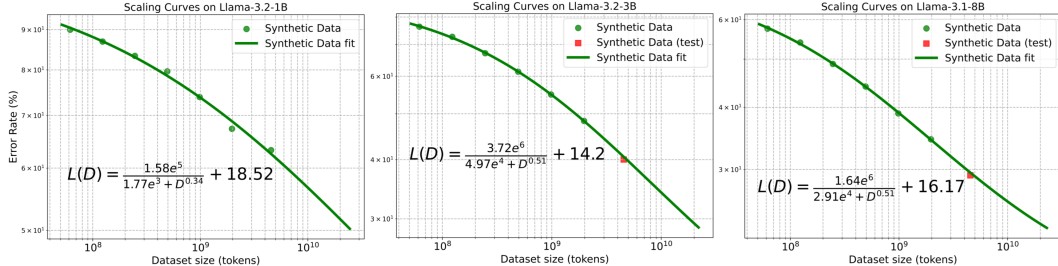

Figure 1: Scaling laws across different model sizes. The x-axis denotes the number of training tokens, while the y-axis shows error rates on the MATH benchmark. Green points indicate data sizes used for fitting the scaling curves, and red points assess the predictive accuracy of these fitted curves.

synthetic token volume. As shown in Figure 1, 1B model exhibits a smaller $\beta$ (0.34) compared to larger models (3B and 8B), suggesting it benefits less from increased synthetic data. The ratio $\frac{B}{D_l}$ correlates strongly with initial model performance, where a smaller ratio implies higher initial capability. The 7B model achieves the lowest ratio (56.3), reflecting its superior model capacity and richer pre-training exposure. Besides, identical decay exponents ($\beta$) for 3B and 7B models indicate that performance differences primarily arise from disparities in initial knowledge and model capacity rather than synthetic data scaling efficacy.

**Predictable Performance** We evaluated the predictive accuracy of our fitted scaling curves for the 3B and 8B models. As shown by the red squares in Figure 1, these curves reliably forecast the models' error rates (40.0% for 3B, 28.7% for 8B) on the MATH benchmark when scaled to 4.5B synthetic tokens, corresponding to accuracies of 60.0% and 71.3%, respectively. These results underscore the effectiveness of scaling synthetic data: even as traditional pre-training data approaches saturation, systematic expansion of synthetic datasets continues to provide stable and predictable performance improvements. Using our scaling curves, we further extrapolate the future performance of the 3B and 8B models as the volume of synthetic data continues to grow. Table 1 presents these projections, indicating diminishing returns in performance improvements beyond approximately 300B synthetic tokens.

Specifically, we estimate the 3B model would require around 4T synthetic tokens to approach its performance ceiling, whereas 8B model would need about 1T tokens. Currently, SYNTHLLM generates only five questions per document from pre-training corpus, which is still far from fully leveraging the potential of each document (see Figure 5). We can easily continue scaling up our synthetic dataset to further approach the performance ceiling.

| Model | 10B tokens | 50B tokens | 250B tokens | 300B tokens | 1T tokens | 4T tokens |
|-------|-----------|-----------|------------|------------|----------|----------|
| 3B | 64.6 | 74.7 | 80.5 | 80.9 | 83.1 | 84.4 |
| 8B | 73.2 | 78.6 | 81.4 | 81.6 | 82.6 | 83.2 |

Table 1: The predicted MATH accuracies (%) on 3B and 8B models when continuing to scale synthetic data based on our scaling law.

## 3 SYNTHLLM: Web-Scale Synthetic Data Generation

In this section, we detail the SYNTHLLM framework for synthetic data generation. For a given target domain (e.g., mathematical reasoning), we first filter high-quality, domain-specific reference documents from Fineweb-Edu (Penedo et al., 2024), a publicly available web data repository. Using these filtered documents, we then generate large-scale, diverse questions (or prompts) by leveraging open-source LLMs through three distinct methods, each designed to progressively enhance question diversity. Finally, we generate corresponding answers to these questions, again utilizing open-source LLMs.

### 3.1 Reference Documents Filtering

**Cold-start Domain Classifier.** For a given target domain, we train a binary classifier to filter web documents from Fineweb-Edu. While negative examples can be randomly sampled, obtaining sufficient positive examples for each domain is more challenging. To address this, we use synthetic documents as positive examples. Specifically, we generate synthetic

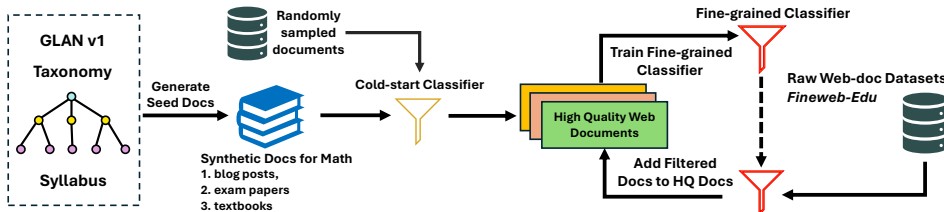

Figure 2: Overview of reference documents filtering.

domain-specific documents based on syllabi from GLAN (Li et al., 2024b), which cover 126 distinct disciplines. Each syllabus is produced by GPT-4 "(almost) from scratch", with further details available in Li et al. (2024b). As illustrated in Figure 2, we first use these syllabi to prompt a large language model (LLM) to generate synthetic documents representative of the target domain (e.g., blog posts, exam papers, and textbooks). Once we obtain the synthetic positive examples and randomly sampled negative examples from Fineweb-Edu, we train a cold-start domain classifier. Applying this classifier yields an initial set of domain-relevant documents, $\mathcal{D}^0$, from Fineweb-Edu.

**Fine-grained Domain Classifier.** The initial reference document set, $\mathcal{D}^0$, is typically small. To expand it, we iteratively classify Fineweb-Edu documents based on previously identified references, $\mathcal{D}^{t-1}$. At each iteration $t$, we randomly sample an additional subset $\mathcal{D}_R^{t-1}$ from Fineweb-Edu and instruct GPT-4o to evaluate all documents in $\mathcal{D}^{t-1} \cup \mathcal{D}_R^{t-1}$ on a 1–10 scale, considering relevance to the target domain, clarity, and language quality. These ratings are then used to train a fine-grained classifier, which we apply to Fineweb-Edu to construct the next iteration's reference set, $\mathcal{D}^t$. We retain documents with scores above a threshold of 6.5, as it yields satisfactory performance in initial experiments. Empirically, two iterations suffice to generate a high-quality and sufficiently large reference corpus.

Both the cold-start and fine-grained classifiers are implemented using random forests, trained on Fineweb-Edu documents represented as vector embeddings generated by StableLM-2-1.6B.

## 3.2 Document-Grounded Question Generation

After applying the filtering process outlined in Section 3.1, we obtain the refined document set $\mathcal{D}$. To generate questions, we employ three distinct methods, utilizing either single-document or multi-document contexts to progressively enhance question diversity. Let $\mathbf{M}^Q$ denote the LLM used for question generation.

**Level-1 Generator.** In the first method, we generate questions from a single reference document $d \in \mathcal{D}$. Since $d$ may already contain questions relevant to the target domain, we first prompt the LLM $\mathbf{M}^Q$ to determine whether $d$ includes any existing questions. If such questions exist, we directly extract them and encourage $\mathbf{M}^Q$ to rephrase them freely for clarity and improved understanding. We also instruct $\mathbf{M}^Q$ to create additional questions inspired by the content of $d$. Finally, $\mathbf{M}^Q$ annotates each question with a special tag, either <original_question> or <newly_created>, to indicate whether the question is an original question from $d$. The detailed prompt for $\mathbf{M}^Q$ is in Figure 7 of Appendix. Similar directly extraction-based synthesis method has been adopted in Yue et al. (2024), Zhou et al. (2024), and Yuan et al. (2025). However, the Level-1 approach described above may be limited in scalability, as the diversity and content of newly generated questions is constrained by the finite set of high-quality reference documents $\mathcal{D}$ containing questions.

**Level-2 Generator.** To overcome limitations of previous approach and better utilize $d$, we draw inspiration from pedagogical principles. In conventional education (Council et al., 2005; Anderson & Krathwohl, 2001), instructors first introduce fundamental knowledge and key concepts before guiding students toward deeper understanding through exercises, fostering problem-solving and critical thinking skills. Our Level-2 method mirrors this approach and implement it in two stages, as illustrated in Figure 3.

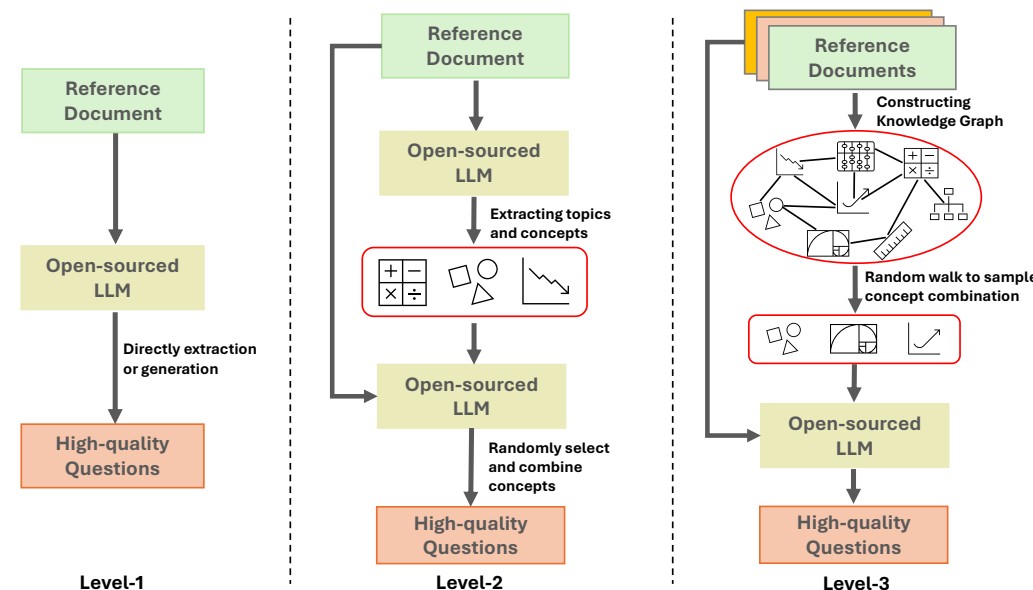

Figure 3: Illustrations of our Document-Grounded Question Generation methods.

First, we prompt the model to act as an instructor, extracting key topics and concepts from $d$ to serve as the foundation for question generation. Specifically, we prompt $\mathbf{M}^Q$ with $\mathbf{p}_k$ to identify relevant *topics* (e.g., "Curvature", "Central Limit Theorem") and *key concepts* (e.g., "Arc length of a curve", "Binomial Approximation to Normal Distribution"), formally defined as: $\mathbf{K} = \mathbf{M}^Q(\mathbf{p}_k, d)$, where $\mathbf{K}$ represents the extracted meta-information. The detailed prompt $\mathbf{p}_k$ is provided in Figure 10, while Figure 11 and 12 illustrate examples of extracted $\mathbf{K}$. In the second stage, we generate questions while introducing controlled randomness to enhance diversity. Specifically, we instruct $\mathbf{M}^Q$ to randomly select and combine concepts from $\mathbf{K}$, forming a subset $\mathbf{k}_s$. The final question is then generated by $\mathbf{M}^Q$, conditioned on both the reference document $d$ and $\mathbf{k}_s$. We aim for the generated questions to remain grounded in the reference document $d$ and strictly adhere to the selected subset $\mathbf{k}_s$. The overall question generation process is formally defined as:

$$\mathbf{x} = \mathbf{M}^Q((\mathbf{p}_q, \mathcal{R}(\mathbf{K}), d)), \quad \mathbf{K} = \mathbf{M}^Q((\mathbf{p}_k, d)), \tag{4}$$

where $\mathbf{k}_s = \mathcal{R}(\mathbf{K})$, $\mathcal{R}$ is random selection operator and $\mathbf{x}$ is the generated questions. For simplicity, we integrate subset selection of $\mathbf{k}_s$ and the generation of the final questions into a single prompt $\mathbf{p}_q$, detailed in Figure 8 of the Appendix. As shown in Eq. (4), $\mathbf{x}$ rely on both (random) concept combinations generated by $\mathcal{R}(\mathbf{K})$ and $d$. This design intuitively offers greater diversity than Level-1 method, which depends solely on $d$. This brings significant advantages. High-quality question synthesis is no longer constrained by the limited existing questions within reference documents. Instead, by decomposing and recombining key concepts, our Level-2 approach more efficiently exploits limited reference documents, thereby enabling more scalable synthetic question generation. In Section 4.3, we empirically validate these benefits through a series of ablation studies. The set of topics and key concepts $\mathbf{K}$ resembles the syllabus from GLAN (Li et al., 2024b), but unlike GLAN, each $\mathbf{K}$ in our approach is explicitly grounded in a reference document $d$.

**Level-3 Generator.** Although Level-2 method enhances question diversity by combining different topics and key concepts within a single document, each document inherently covers only a limited scope of material. To further improve diversity and scalability, the Level-3 generator extends the Level-2 approach by incorporating concepts from multiple documents and establishing cross-document grounding. The process is illustrated in Figure 3.

**Global Concept Graph Construction.** In the Level-2 method, topics and key concepts (i.e., $\mathbf{K}$) are extracted from a *single* document. Here, we extend this modeling to a global level

by capturing relationships among all high-level concepts across documents. Specifically, we construct a graph $G$, where nodes represent all distinct topics $\mathbb{T}$ and key concepts (KCs) $\mathbb{C}$. The intuition behind this construction is that topics and key concepts frequently co-occurring within the same document likely indicate meaningful associations. Thus, we define edge weights based on co-occurrence statistics. The graph consists of three types of edges: topic-topic edges, topic-KC edges, and KC-KC edges, forming three subgraphs: the topic graph, the topic-KC graph, and the KC graph. An edge is established between two nodes, $\mathbf{u}$ and $\mathbf{v}$, whenever they co-occur in the same document. The edge weight $w(\mathbf{u}, \mathbf{v})$ is defined as:

$$w(\mathbf{u}, \mathbf{v}) = \log\left(\text{freq}(\mathbf{u}, \mathbf{v}) + \epsilon\right) \tag{5}$$

where $\text{freq}(\mathbf{u}, \mathbf{v})$ denotes the raw co-occurrence count of $\mathbf{u}$ and $\mathbf{v}$, and $\epsilon$ is a small positive constant for numerical stability.

**Concept Combination Sampling.** The sampling process begins with uniform random sampling from the nodes in the topic sub-graph $\mathbb{T}$. We iterate through all topics in $\mathbb{T}$ over multiple epochs. After selecting an initial topic at random, we perform a random walk of one to two steps within the topic sub-graph to identify additional related topics. The transition probability from node $\mathbf{u}$ to node $\mathbf{v}$ is computed as:

$$p_{\mathbf{u}, \mathbf{v}} = \frac{\exp\left(w(\mathbf{u}, \mathbf{v})\right)}{\sum_{\mathbf{v}' \in \mathcal{N}(\mathbf{u})} \exp\left(w(\mathbf{u}, \mathbf{v}')\right)} \tag{6}$$

where $\mathcal{N}(\mathbf{u})$ denotes the set of adjacent nodes to $\mathbf{u}$ in the topic sub-graph. Next, we sample key concepts corresponding to the selected topics. We begin with a one-step random walk on the topic-KC sub-graph, using transition probabilities from Equation (6). Additional related key concepts are then sampled through three to four steps of random walks within the KC sub-graph, resulting in a sampled set $\mathbf{K}^g$. Since $\mathbf{K}^g$ serves as a high-level summary, certain details from the original reference documents may be omitted. Thus, we search for reference documents most aligned with $\mathbf{K}^g$ as grounding. Specifically, we compute the Jaccard similarity between $\mathbf{K}^g$ and the concept sets of each document in $\mathcal{D}$, selecting top two documents with the highest similarity scores as references. Following the Level-2 procedure, we instruct $\mathbf{M}^Q$ to randomly select and combine concepts from $\mathbf{K}^g$, using the selected concepts along with the two reference documents for question generation. The detailed prompt is provided in Figure 9 of Appendix. Unlike Level-2, which is limited to a *single* document, Level-3 grounds on multiple documents and employs concept sampling from a global graph, significantly enhancing the diversity and breadth of generated questions.

### 3.3 Answer Generation

Using the methods described in Section 3.2, we obtain a large set of questions and proceed with answer generation. We employ open-source LLMs, such as Qwen2.5-Math-72B-Instruct, as our answer generator $\mathbf{M}^A$. Currently, we do not incorporate an additional answer verification process, as preliminary experiments indicate that our synthetic data already achieves satisfactory quality. The detailed experimental results are shown in Section C.3. However, further improvements could be made by refining answer validation through techniques like majority voting (Wang et al., 2022; Jiao et al., 2024) or multi-agent debate (Du et al., 2024; Khan et al., 2024), albeit at additional inference computational cost. We leave this for future work.

## 4 Experiments

### 4.1 Experimental Settings

**Models and Our Datasets.** For $\mathbf{M}^Q$, we use Mistral-Large-Instruct-2407 (team, 2024), as larger models inherently possess the internal knowledge necessary for effective concept extraction and question generation. For $\mathbf{M}^A$, we employ Qwen2.5-Math-72B (Yang et al., 2024). The trained models are Llama-3.2-1B, -3B, and Llama-3.1-8B base models (Llama Team, 2024). After filtering, we collected approximately 520K math documents. For Level-1 and Level-2, we generated 5 questions per reference document, yielding approximately 2.3M and

| Model | GSM8K | MATH | Minerva | Olympiad | College | Gaokao | Average |
|---|---|---|---|---|---|---|---|
| Llama-3.2-1B-Instruct | 47.2 | 28.0 | 5.9 | 5.5 | 18.8 | 25.2 | 21.8 |
| SYNTHLLM-1B (3.2M) | 45.7 | 32.9 | 6.6 | 7.6 | 27.3 | 28.8 | 24.8 |
| SYNTHLLM-1B (7.4M) | **50.4** | **37.4** | **8.1** | **8.3** | **31.7** | **30.6** | **27.4** |
| Llama-3.2-3B-Instruct | 78.0 | 47.5 | 17.3 | 14.8 | 31.8 | 37.7 | 37.9 |
| SYNTHLLM-3B (3.2M) | 78.1 | 54.3 | 16.9 | 17.5 | 38.3 | 46.5 | 41.9 |
| SYNTHLLM-3B (7.4M) | **80.7** | **60.0** | **18.8** | **21.9** | **42.3** | **50.9** | **45.8** |
| Llama-3.1-8B-Instruct | 84.2 | 48.9 | 25.7 | 13.2 | 32.1 | 43.4 | 41.2 |
| Llama-3.1-70B-Instruct | **94.5** | 66.1 | **34.2** | 29.6 | 41.4 | 56.6 | 54.2 |
| NuminaMath-CoT-7B | 75.4 | 55.2 | 19.1 | 19.9 | 36.9 | 47.5 | 42.3 |
| NuminaMath-CoT-72B | 90.8 | 66.7 | 25.0 | 32.6 | 39.7 | 54.0 | 51.5 |
| MAmmoTH2-Plus-8B (10M) | 78.4 | 41.9 | 10.7 | 11.3 | 16.1 | 31.9 | 31.7 |
| JiuZhang3.0-8B (6M) | 88.7 | 51.2 | 21.7 | 18.8 | 37.5 | 43.4 | 43.6 |
| NaturalReasoning-8B (2.8M)* | - | 55.6 | - | - | - | - | - |
| OpenMathInstruct-2-8B (14M) | 91.1 | 67.5 | 22.5 | 27.7 | 39.2 | 53.5 | 50.3 |
| SYNTHLLM-8B (3.2M) | 88.4 | 66.1 | 25.4 | 30.2 | 44.3 | 56.9 | 51.9 |
| SYNTHLLM-8B (7.4M) | 92.1 | **71.3** | 26.5 | **33.0** | **45.3** | **61.0** | **54.9** |

Table 2: Performance of models of different sizes on various math benchmarks. All metrics are reported as percentages (%). We evaluate three model sizes, with the best result highlighted in **bolded**. The number in () represents training sample number. NaturalReasoning-8B: Results are directly referenced from the original paper, as the full dataset and model have not been released.

2.6M samples, respectively. For Level-3, each sampled $\mathbf{K}^g$ generated 3 questions, resulting in 2.6M samples. Further details of training and dataset are provided in Section B.

**Other Baselines and Evaluation Benchmarks.** We compare our approach with MAm-moTH2 (Yue et al., 2024), JiuZhang3.0 (Zhou et al., 2024), NaturalReasoning (Yuan et al., 2025), and OpenMathInstruct-2 (Toshniwal et al., 2025). Additionally, we evaluate our Level-2 and Level-3 methods against existing augmentation techniques aimed at enhancing question diversity. Specifically, we consider two widely used approaches: *rephrase augmentation* (Yu et al., 2024) and *persona augmentation* (Ge et al., 2024; Lambert et al., 2024; Wang et al., 2025), both of which generate new questions based on those extracted in Level-1. Following Qwen-Math (Yang et al., 2024), we adopt evaluation benchmark: MATH (Hendrycks et al., 2021), GSM8K (Cobbe et al., 2021), OlympiadBench-Math (Olympiad) (He et al., 2024), CollegeMath (College) (Tang et al.), Minerva Math (Minerva) (Lewkowycz et al., 2022), and Gaokao 2023 En (Gaokao) (Liao et al., 2024). The more details are shown in Section B.

## 4.2 Main Experimental Results

We evaluate SYNTHLLM across multiple mathematical reasoning benchmarks of varying difficulty levels. Table 2 presents results, with values in parentheses indicating sample sizes. For baseline datasets, we directly evaluate their released models. We train models using both the full SYNTHLLM dataset (7.4M samples) and a subset obtained by uniformly sampling 3.2M examples, denoted as SYNTHLLM-1B, SYNTHLLM-3B, and SYNTHLLM-8B.

**SYNTHLLM Outperforms Other Synthetic Datasets.** SYNTHLLM consistently surpasses other synthetic datasets across all benchmarks, demonstrating the effectiveness of SYNTH-LLM. OpenMathInstruct-2 is a math dataset curated from MATH and GSM8K. Despite its larger size, the model trained on smaller dataset (SYNTHLLM-8B (3.2M)) achieves comparable performance on in-distribution test sets (MATH and GSM8K) while significantly outperforming it on OOD benchmarks, highlighting the superior generalization of SYNTH-LLM. Additionally, our models consistently surpass Llama Instruct models of similar sizes and even match or outperform much larger models, such as NuminaMath-CoT-72B and Llama-3.1-70B-Instruct. Notably, we currently generate only five questions per document for Level-1 and Level-2 and three questions for Level-3. As shown in Figure 5, this is far from fully utilizing each document's potential. Simply increasing the number of synthesized questions per document could further enhance benchmark performance.

## 4.3 Ablation Studies about Our SYNTHLLM Framework

**Comparison with Direct Extraction-Based Synthesis.** We compare Level-1 (direct extraction-based synthesis) with Level-2 and Level-3 across models of different sizes (1B and

3B). As shown in Table 4 of Appendix, on the 3B model, Level-2 and Level-3 significantly outperform Level-1 and the instruct-tuned version across all benchmarks. On the 1B model, Level-3 achieves substantial gains over Level-1 on most benchmarks, except for College and Gaokao. While each document yields five sampled questions, the final dataset size for Level-1 (2.3M) is notably smaller than that of Level-2 (2.6M). This is due to the high redundancy in direct extraction, where many duplicate questions are generated and later removed. This highlights the scalability limitations of direct extraction-based approach.

**Diversity of Generated Questions.** Compared to direct extraction-based synthesis, SYNTHLLM generates more diverse questions by decomposing and recombining knowledge concepts. To assess this, we compare Level-1 and Level-2 by measuring the similarity among generated questions within the same document. Specifically, we randomly select 2,000 documents and generate 50 questions per document for each method.

We then use all-MiniLM-L6-v2 (Wang et al., 2020) to obtain question embeddings and compute cosine similarity between each question pair from same $d$. The histogram of question similarity is shown in Figure 4. The results clearly indicate that Level-2 generates more diverse questions, as evidenced by the lower similarity scores among questions derived from same $d$.

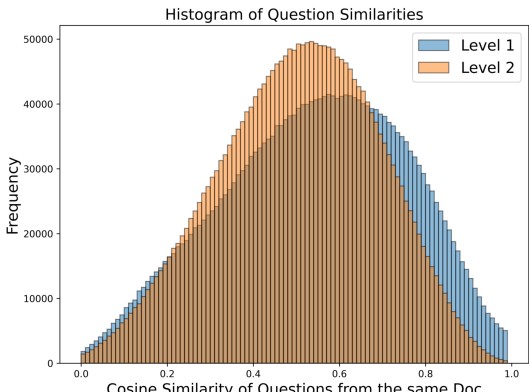

Figure 4: Histogram of question similarity within the same document. The x-axis represents the cosine similarity between question pairs from the same document, while the y-axis denotes the frequency of each bin.

**Comparison with data augmentation methods on limited reference documents.** We compare our Level-2 and Level-3 methods with augmentation techniques, including rephrase (Yu et al., 2024) and persona augmentation (Ge et al., 2024). We fix number of reference documents and evaluate different methods for scaling high-quality question generation from it. The effectiveness of each method will be measured by its ability to consistently improve model performance on downstream tasks as the dataset expands. Specifically, we randomly select 2,000 reference documents and generate 25, 50, to 150 questions per document using Level-2 and Level-3 methods, resulting in datasets ranging from 50K, 100K, to 300K questions. For rephrasing and persona augmentation, we need seed questions to conduct data augmentation. Therefore, we first adopt our Level-1 method to extract existing questions from the 2,000 fixed reference documents (yielding 6,500 questions) as the seed dataset. For above questions, we employ the same answer generator $\mathbf{M}^A$ (see Section 4.1). We train these datasets on the Llama-3.1-8B and present the results in Figure 5, where (a) shows MATH accuracy and (b) depicts average accuracy across benchmarks. Level-2 and Level-3 consistently outperform augmentation-based methods. Notably, as the number of generated questions per document increases to 150, our methods continue to yield performance gains, whereas rephrase and persona augmentation reach saturation. This highlights our approaches can efficiently utilize limited reference documents by decomposing and recombining knowledge concepts, enabling more scalable and high-quality question generation. In previous experiments, we generated only 5 questions per document for Level-2 and 3 questions per document for Level-3. These results suggest that our approach has not yet reached its full potential in document utilization and can be further expanded.

## 4.4 Preliminary Experiments in Coding Domain

To explore the generalizability and effectiveness of our method, we conduct preliminary experiments in the coding domain. We adopt the same SYNTHLLM framework (described in Section 3) and follow the experimental setup outlined in Section 4.1. Specifically, we generate 2.2M samples using our Level-1 baseline method, and 2.4M and 2.5M samples

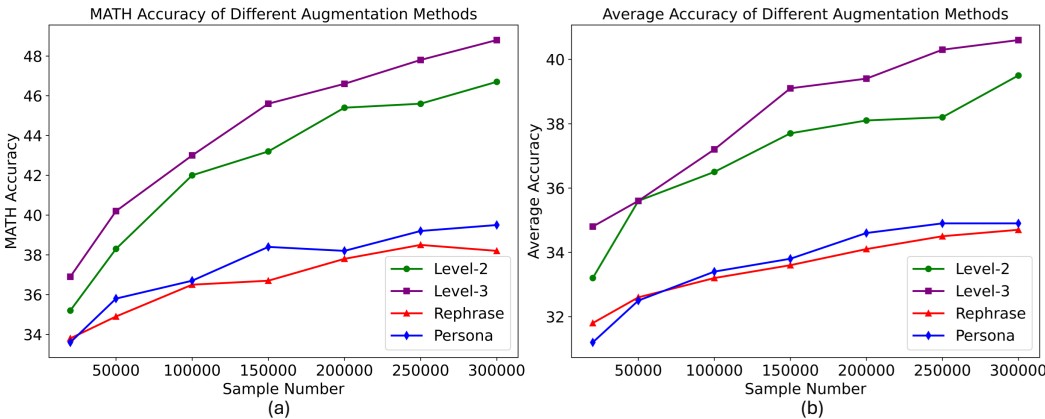

Figure 5: **(a)**: The performances of Level-2, Level-3, and other augmentation methods on MATH benchmark. **(b)**: The average performances across various benchmarks. The x axis denotes the sample number. The y axis represents the accuracy (%).

for our Level-2 and Level-3 methods, respectively. Mistral-Large-Instruct-2407 is used as the question generator, while Qwen2.5-Coder-32B-Instruct serves as the answer generator.

We then train the Llama-3.1-8B base model on these datasets. The results on HumanEval (Chen et al., 2021) and MBPP (Austin et al., 2021) are summarized in Table 3. For comparison, we also include the performance of the instruction-tuned model (Llama-3.1-8B-Instruct) and a code-specific model (Deepseek-Coder-v1.5-Instruct (Guo et al., 2024)).

| Model | HumanEval | MBPP |
|---|---|---|
| Llama-3.1-8B-Instruct | 72.6 | 71.2 |
| Deepseek-Coder-v1.5-Instruct | 75.6 | 73.4 |
| SYNTHLLM Level-1 | 73.2 | 67.3 |
| SYNTHLLM Level-123 | 78.7 | 72.5 |

Table 3: HumanEval and MBPP Performance (%)

SYNTHLLM Level-1 represents the model trained solely on Level-1 data, while the Level-123 model denotes the model trained on the entire synthetic dataset (combined data from 3 methods). Our newly proposed Level-2 and Level-3 methods yield significant improvements in coding performance over the Level-1 baseline. SYNTHLLM Level-123 outperforms the instruction-tuned model and achieves performance comparable to the code-specific model, demonstrating the generalizability and effectiveness of our approach in coding.

## 5   Conclusions and Future Work

We investigate scaling laws of synthetic data by introducing SYNTHLLM. Our findings show that synthetic data generated by SYNTHLLM follows a rectified scaling law, enabling accurate performance prediction when doubling the dataset size. This suggests that systematically expanding the synthetic dataset yields sustained and predictable performance gains. Comparisons with existing synthetic data generation and augmentation methods demonstrate that SYNTHLLM offers superior performance and scalability. Our framework could extend to domains such as physics, law, and finance. Future directions include refining SYNTHLLM to optimize pre-training data utilization and answer verification for improved accuracy.

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

## A   Related Work

**Synthetic Data Generation.** Synthetic data has emerged as a promising method to mitigate data scarcity and reduce the costs associated with data collection and annotation. *In this work, we focus specifically on synthetic data generation for the post-training phase.* Conventional methods typically leverage a limited number of high-quality human-annotated seed examples to generate diverse augmentations via large language models (LLMs). Common techniques include: 1) sampling seed instructions as few-shot prompts for generating new instructions (Wang et al., 2022; Honovich et al., 2022; Toshniwal et al., 2025; Li et al., 2024a); and 2) rephrasing seed samples to create variations (Yu et al., 2024; Maini et al., 2024; Ge et al., 2024; Lambert et al., 2024; Luo et al., 2023). However, reliance on scarce high-quality seeds limits scalability and diversity. Recent approaches by Xu et al. (2024) and Li et al. (2024b) propose generating synthetic data from scratch by exploiting either LLM uncertainty or predefined knowledge taxonomies. In contrast, pre-training data—vast and inherently diverse—remains underutilized for scalable post-training synthetic data generation. Prior methods have either directly extracted samples from web documents (similar to our Level-1 approach) (Yue et al., 2024; Zhou et al., 2024; Yuan et al., 2025) or used document back-translation (Li et al., 2024). Our proposed SYNTHLLM method aligns with this research direction but provides a more efficient framework for leveraging web documents to scale the diversity of synthetic data.

**Scaling Law of LLMs.** Scaling laws predict how model performance varies with model size and data volume, providing insights critical for optimizing computational resources during model training (Hestness et al., 2017; Rosenfeld et al., 2019; Kaplan et al., 2020; Hoffmann et al., 2022; Bahri et al., 2024). Recent advancements have introduced refined scaling laws, such as data-constrained scaling (Muennighoff et al., 2023), hyperparameter scaling Bi et al. (2024), and model distillation Busbridge et al. (2025). For fine-tuning LLMs, Hernandez et al. (2021) explored scaling behavior transitioning from unsupervised pre-training, emphasizing pre-training's critical role. Lin et al. (2024) further introduced a rectified scaling law tailored specifically for fine-tuning tasks. Our work extends this rectified scaling law (Lin et al., 2024), demonstrating for the first time that scaling laws also apply robustly when fine-tuning LLMs using synthetic data.

## B   Details of Training Model and Our Dataset

**Training Settings.**   We employ Supervised Fine-Tuning (SFT) to train the model. We adopt AdamW optimizer (Loshchilov & Hutter, 2019) and set learning rate to $1 \times 10^{-5}$ with a batch size of 512. We utilize the linear learning rate scheduler and train the model for 3 epochs.

**Our Dataset.**   In this work, we focus on the math domain. After filtering high-quality documents from web data, we collect approximately 520K math-related documents. For Level-1 and Level-2, we generate 5 questions per document, followed by *deduplication* and *decontamination* (Yue et al., 2024; Toshniwal et al., 2025) using evaluation benchmarks, yielding around 2.3M and 2.6M samples, respectively. For Level-3, the constructed concept graph contains around $32\,000$ topics and $200\,000$ knowledge concepts. We conduct random walks on the topic sub-graph for five epochs, covering each topic in each epoch. For each sampled $\mathbf{K}^g$, we generate 3 questions, producing about 2.6M samples. Due to the high computational cost of training on large-scale datasets, we limit the number of sampled questions per document in Level-2 and constrain the number of random walk epochs in Level-3 to a relatively small scale (e.g., five epochs). It is still far from fully leveraging the potential of each document (see Figure 5). For $\mathbf{M}^Q$, the output temperature is 0.75. For $\mathbf{M}^Q$, the output temperature of $\mathbf{M}^A$ is 0.

**Statistics of Synthetic Questions and Answers.**   The median lengths of the questions generated by Level-1, Level-2, and Level-3 are 32, 66, and 80, respectively, while the corresponding median lengths of the answers are 358, 497, and 545.

**Other Baselines.** We compare our synthetic data with other high-quality synthetic datasets. MAmmoTH2 (Yue et al., 2024), NaturalReasoning (Yuan et al., 2025), and JiuZhang3.0 (Zhou et al., 2024) employ a direct extraction-based synthesis approach on pre-training documents, similar to our Level-1 method. OpenMathInstruct-2 (Toshniwal et al., 2025) is a large-scale synthetic math dataset consisting of 14M samples, with GSM8K and MATH serving as the seed datasets. Additionally, we compare our dataset with NuminaMath (LI et al., 2024), a high-quality dataset that includes both synthetically generated data and extracted question-answer pairs from exam papers and mathematics discussion forums.

Furthermore, we evaluate our Level-2 and Level-3 approaches against existing methods designed to enhance question diversity through augmentation. Specifically, we include two popular augmentation approaches: the *rephrase augmentation* approach (Yu et al., 2024; Maini et al., 2024) and the *persona augmentation* approach (Ge et al., 2024; Lambert et al., 2024), both of which generate new questions based on the questions extracted using Level-1. For rephrase augmentation, we follow the same procedure as MetaMath (Yu et al., 2024), providing two-shot examples in the prompt to rephrase questions. For persona augmentation, we utilize the 200K personas released by the authors as prompts for question augmentation.

**Evaluation.** To evaluate the math reasoning ability of our model trained on synthetic data, we utilize the evaluation code from Qwen-Math (Yang et al., 2024) to conduct assessments on several mainstream benchmarks: MATH (Hendrycks et al., 2021), GSM8K (Cobbe et al., 2021), OlympiadBench-Math (Olympiad) (He et al., 2024), CollegeMath (College) (Tang et al.), Minerva Math (Minerva) (Lewkowycz et al., 2022), and Gaokao 2023 En (Gaokao) (Liao et al., 2024). The output temperature for all tested models is set to 0.

**Computational Cost.** For generating questions, answers, and key concepts, we utilized a total of 32 Nvidia H100 GPUs, with each inference task running on 4 Nvidia H100 GPUs. The complete dataset generation process took approximately 3 days. For model training, we employed AMD MI300 GPUs; the largest training task utilized 32 AMD MI300 GPUs and required around 2 days to complete.

## C   More Results

In this section, we present additional experimental results about Section 4.

### C.1   Ablation Study about Our Three Distinct Methods

We compare Level-1 (direct extraction-based synthesis) with Level-2 and Level-3 across models of different sizes (1B and 3B). As shown in Table 2 (Appendix), on the 3B model, Level-2 and Level-3 consistently outperform both Level-1 and the instruct-tuned version across all benchmarks. On the 1B model, Level-3 achieves substantial improvements over Level-1 on most benchmarks, except for College and Gaokao. Although each document contributes five sampled questions, the final dataset size for Level-1 (2.3M) is significantly smaller than that of Level-2 (2.6M). This discrepancy arises from the high redundancy in direct extraction, where many duplicate questions are generated and subsequently removed. These findings highlight the scalability limitations of direct extraction-based synthesis.

### C.2   Ablation Study on Scaling Law Form

We compare the rectified form Eq. (2) and marginal form Eq. (3) in terms of their effectiveness in fitting the scaling curve. The results of fitting scaling curves using these two formulations on the Llama-3.2-3B model are presented in Figure 6. The black dotted line represents the scaling curve fitted using the marginal form (Eq.(3)), while the green solid line corresponds to the fit using the rectified form (Eq.(2)). The green points indicate the data sizes used for fitting the scaling laws, whereas the red points are used to evaluate the predictive performance of the fitted curves. From the results, we observe that the marginal form fails to accurately fit the scaling curve and struggles to predict performance when scaling up

| Model | GSM8K | MATH | Minerva | Olympiad | College | Gaokao | Average |
|---|---|---|---|---|---|---|---|
| Llama-3.2-1B-Instruct | **47.2** | 28.0 | 5.9 | 5.5 | 18.8 | 25.2 | 21.8 |
| Level-1-1B (2.3M) | 38.9 | 28.5 | 6.6 | 5.3 | **26.5** | **27.0** | 22.2 |
| Level-2-1B (2.6M) | 40.1 | 27.2 | 4.8 | 5.5 | 23.3 | 24.9 | 21.0 |
| Level-3-1B (2.6M) | 42.6 | **30.1** | **9.2** | **7.1** | 25.0 | 26.2 | **23.4** |
| Llama-3.2-3B-Instruct | **78.0** | 47.5 | 17.3 | 14.8 | 31.8 | 37.7 | 37.9 |
| Level-1-3B (2.3M) | 71.8 | 46.8 | 16.2 | 13.6 | 36.3 | 42.3 | 37.8 |
| Level-2-3B (2.6M) | 74.1 | **49.9** | 17.2 | 15.3 | **37.4** | **43.4** | 39.6 |
| Level-3-3B (2.6M) | 77.0 | 49.1 | **18.0** | **17.2** | **37.4** | 42.9 | **40.3** |

Table 4: Performance of 1B and 3B models on various math benchmarks. The number in () represents the sample size. The best result is in **bolded**. All metrics are measured in percentage (%).

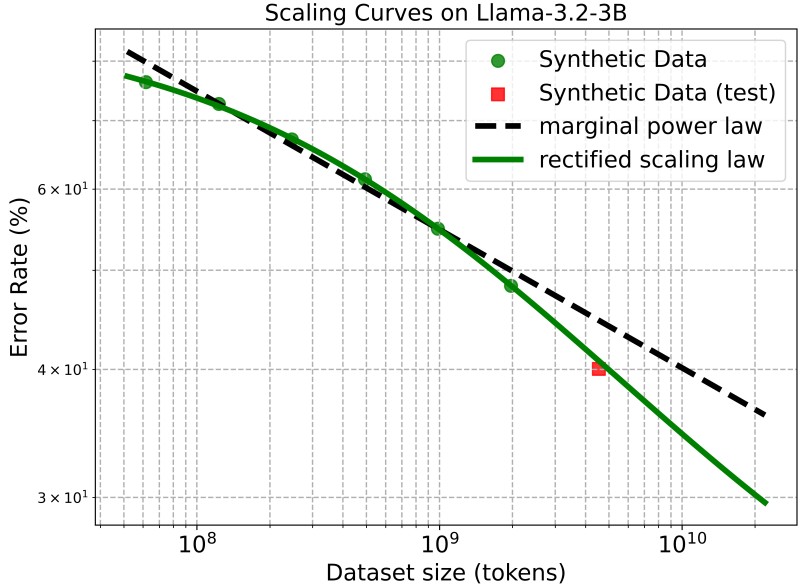

Figure 6: The results of fitting scaling curves using different formulations on the Llama-3.2-3B model. The x axis denotes the number of training tokens. The y axis represents the models' error rates on MATH benchmark. The green points represent the data sizes used to fit the scaling laws, while the red points are used to test the prediction performance of the fitted curves.

the synthetic dataset. In contrast, the rectified form effectively captures the scaling law while also demonstrating strong predictive accuracy when scaling up the synthetic dataset. This underscores the importance of introducing the pre-learned size $D_l$ for scaling law of fine-tuning LLMs.

## C.3 Ablation Study on Answer Verification Process

The decision not to conduct further answer verification was motivated by our prior preliminary experiments, which showed that removing noisy samples during SFT yielded only marginal gains in model performance. We describe these experiments below. Our preliminary experiments followed the same experimental setup as the ablation studies described in Section 4.3. We randomly selected 2,000 reference documents and generated 50 questions per document using our Level-2 method. For each question, we generated 8 candidate answers.

For each question, we used Llama-3.1-70B-Instruct as an answer evaluator to score its 8 candidate answers and discarded any judged "potentially incorrect". We then randomly picked one answer from the remaining candidates. Without filtering, one answer was randomly selected from all 8 candidates. In both cases, only one answer per question was retained for comparison. We then trained the Llama-3.1-8B

| Model | MATH | Average |
|---|---|---|
| Llama-3.1-8B wo filtering | 42.0 | 36.5 |
| Llama-3.1-8B w filtering | 42.2 | 36.6 |

Table 5: MATH and Average Performance (%)

base model on these generated datasets. We evaluate each trained model on the MATH benchmark and compute the average accuracy across all other benchmarks. As shown Table 5, filtering out noisy samples results in only marginal gains in model performance.

Recent studies (Tang et al.; Li et al., 2025; Gandhi et al., 2025) report consistent findings. As shown in Table 7 of (Tang et al.), adding a validation step for synthetic data does not improve performance on mathematical reasoning tasks. Table 2 of (Li et al., 2025) and Figure 6 of (Gandhi et al., 2025) further show that models trained even on entirely incorrect answers achieve performance comparable to those trained on correct answers. These results, along with our experiments, suggest that for SFT, model performance remains relatively robust to a certain degree of noise in answer correctness.

## D    Used Prompts

In this section, we present prompts used in our SYNTHLLM framework.

Here is an article crawl from the web, which our classifier has identified as having significant educational value for students learning math.
As a senior **math** instructor, your task is to create **challenging computation-based math questions**. These questions should be suitable for various contexts, such as homework assignments, exams, interview preparations, classroom activities, competitions, and tutoring sessions while enhancing students' reasoning and critical-thinking skills. Ensure that questions are **non-redundant**, precise, and engaging.

### Guidelines for Creating Computation-based Questions:
1. **Assess Suitability**: If this article does not contain math-related content that can be used to generate engaging and solvable questions, please directly output "NOT SUITABLE for creating questions."
2. **Generate Questions**: If the article is suitable for creating math questions, generate **1 to 5** questions based on the richness and depth of the article content. For articles covering multiple topics, aim to generate more questions to ensure coverage:
    - Each question must be solvable independently and should not rely on
    answers from previous questions.
    - If a question closely resembles the original text, append "<original_question>" at
    the end of the question.
    - If a question is newly created, append "<newly_created>" at the end of the question.
3. **Content Alignment**: Your math questions must exclusively draw from the content of the article, ensuring they are directly aligned with the concepts presented.
4. **Use Original Questions**: Use original questions from the article whenever possible. However, feel free to rephrase them for clarity and improved understanding.
5. **Create New Questions**: Attempt to formulate **newly diverse and challenging questions** that explore different aspects of the content presented in the article.
6. **Self-Contained Questions**: Ensure that each question is self-contained, meaning students do not need to read the article to answer them.
7. **Logical Consistency**: Verify that the questions are **logically sound and directly aligned with the mathematical principles** in the article. You MUST minimize the use of **sub-questions**, unless they are essential to the problem's complexity. 8. **Clarity and Precision**:
    - Use precise and unambiguous language.
    - Write all mathematical expressions or formulas in LaTeX for clarity.
    - Clearly state all assumptions or conditions.
    - The answer should either be exact, or if not possible, then the question should clearly say the
    answer is only expected to be approximately correct.

#### Article
{{ text }}

#### Output Format For each question, provide the following information:
- **Question Content**: The actual math question.
- **School Level**: Specify the school level (e.g., <elementary>, <middle_school>, <high_school>, <college>, <grad_school>, <competition>).
- **Originality Tag**: Append "<original_question>" for original questions from the article or"<newly_created>" for newly created questions.

Example Output:
<Q1>
Question: Content
Orig_tag:<original_question>
Level:<high_school>
</Q1>
<Q2>
Question: Content
Orig_tag:<newly_created>
Level:<college>
</Q2>

Figure 7: Prompt for Level-1. The {{ text }} is the placeholder for inserting the reference document.

As a senior **math** instructor, your task is to create **diverse and challenging computation-based math questions**. These questions should demonstrate the application of the provided topics and key concepts while enhancing students' reasoning and critical-thinking skills. Ensure that questions are **non-redundant**, precise, and engaging.

### Guidelines for Creating Diverse and Challenging Computation-based Questions:
1. **Concept Selection**:
    - Randomly select **up to 2-3 distinct key concepts** from the provided list for each question.
    - Ensure **broad coverage** of the provided concepts across the generated questions,
     avoiding over-relianceon a limited subset of concepts.
    - Avoid repeating the same **concept combinations** or **computational approach**
    across questions.
2. **Diversity and Challenge**:
    - Ensure that each question explores **different combinations of key concepts** and is **sufficiently
    challenging** (e.g., requiring multi-step computations, integrating real-world scenarios,
    involving abstract or advanced reasoning.).
3. **Clarity and Precision**:
    - Verify that the questions are **logically sound**.
    - Use precise and unambiguous language.
    - Write all mathematical expressions or formulas in LaTeX for clarity.
    - Clearly state all assumptions or conditions.
4. **Reference Material**:
    - Use the provided **reference article** as a source of inspiration for generating **unique, diverse,
    and challenging questions**.
    - The reference material is intended to:
      - Supplement the concept list by introducing **novel perspectives**, **contexts**,
      or **applications**.
      - Help create questions that are **more complex, realistic, or uncommon** in
      traditional teaching scenarios.
      - Serve as a resource to craft **real-world scenarios** or **abstract extensions** beyond
      the given concepts.
5. **Output Diversity**:
    - Create between **1 to 5 questions**.
    - Ensure each question is unique in **structure**, **approach**, and **concept usage**.
    - Minimize the use of **sub-questions**, unless they are essential to the problem's complexity.
    - The answer should either be exact, or if not possible, then the question should clearly say
    the answer is only expected to be approximately correct.

### Inputs:
- **Article**:
    {{ text }}
- **Concept List**:
    {{ concept }}

#### Output Format:
<Q1>
Selected Concepts: [Only insert 2-3 concepts here]
Question: [Only insert question here]
</Q1>
<Q2>
Selected Concepts: [Only insert 2-3 concepts here]
Question: [Only insert question here]
</Q2>

Figure 8: Prompt for Level-2. The {{ text }} and {{ concept }} are the placeholders for inserting the reference document and meta-information (e.g., topics, key concepts) extracted from document.

As a senior **math** instructor, your task is to create **diverse and challenging computation-based math questions** based on provided topics and knowledge points. These questions should demonstrate the application of the provided topics and key concepts while enhancing students' reasoning and critical-thinking skills. Ensure that questions are **non-redundant**, precise, and engaging.

You will be provided with a list of key mathematical concepts spanning various topics and two relevant reference materials.

### Guidelines for Creating Diverse and Challenging Computation-based Questions:
1. **Concept Selection**:
    - Adhere to the Provided Topics: Ensure that each question aligns closely with the given topics.
    - Incorporate Multiple Concepts about Different Topics: Each question should encompass **2 or 3 key concepts about different math topics**.
    - Ensure **broad coverage** of the provided concepts across the generated questions, avoiding **over-reliance** on simple or common applications of concepts.
    - Avoid repeating the same **concept combinations** or **computational approach** across questions.
2. **Diversity and Challenge**:
    - Encourage **Cross-Topic Thinking**: By integrating concepts about different math topics, questions will promote holistic understanding and application of mathematical principles.
    - **Leverage the Two Reference Materials**: The combination of both reference materials provides a **broader and more diverse context**, allowing for the creation of questions that explore a wider range of scenarios and applications. Use this to generate questions that challenge students in both familiar and novel contexts.
    - Ensure questions explore **different perspectives** and **applications** of the key concepts. Ensure each question is **sufficiently challenging** (e.g., requiring multi-step computations, integrating real-world scenarios, involving abstract or advanced reasoning.).
3. **Clarity and Precision**:
    - Use precise and unambiguous language.
    - Write all mathematical expressions or formulas in LaTeX for clarity.
    - Clearly state all assumptions or conditions.
4. **Reference Material**:
    - Use the provided **reference articles about different topics** as sources of inspiration for generating **unique, diverse, and challenging questions**.
    - The combination of these two materials allows you to create questions with **more varied perspectives, contexts, and applications**, which can help test students' abilities to apply concepts in different situations.
    - The reference material is intended to:
        - Supplement the concept list by introducing **novel perspectives**, **contexts**, or **applications**.
        - Help create questions that are **more complex, much harder, and uncommon** in traditional teaching scenarios.
        - Serve as a resource to craft **real-world scenarios** or **abstract extensions** beyond the given concepts.
5. **Output Diversity**:
    - Create between **1 to 3 questions**.
    - Ensure each question is unique in **structure**, **approach**, and **concept usage**.
    - Minimize the use of **sub-questions**, unless they are essential to the problem's complexity.
    - The answer should either be exact, or if not possible, then the question should clearly say the answer is only expected to be approximately correct.

### Inputs:
- **Article**:
    {{ text }}
- **Concept List**:
    {{ concept }}

#### Output Format:
<Q1>
Selected Concepts: [Only insert 2-3 concepts here]
Question: [Only insert question here]
</Q1>
<Q2>
Selected Concepts: [Only insert 2-3 concepts here]
Question: [Only insert question here]
</Q2>

Figure 9: Prompt for Level-3. The {{ text }} and {{ concept }} are the placeholders for inserting reference documents and meta-information sampled from knowledge graph.

Here is an article crawl from the web, which our classifier has identified as having significant educational value for students learning math.
Your task is to analyze this article and extract educational materials, specifically focusing on topics and key concepts that can enhance students' understanding of mathematics and improve their problem-solving skills.

Pay special attention to uncommon but important mathematical concepts that are crucial for a deeper understanding.

## Tasks
1. **Determine Educational Level:**
    - Identify the appropriate educational level for the article based on its content. Choose from the following options:
        - Primary School
        - Middle School
        - High School
        - College
        - Graduate School
        - Competition
        - Other
2. **Identify Subject Area:**
    - Specify the primary subject area of mathematics to which the article belongs (e.g., Calculus, Geometry, Algebra, etc.).
3. **Extract Topics and Key Concepts:**
    - **Topics:**
        - List **1 to 5** main topics covered in the article.
        - Use terms commonly recognized in academia or industry.
    - **Key Concepts:**
        - For each identified topic, list **5 to 20** related key concepts.
        - Ensure these concepts are clearly articulated using standard academic or industry terms.

## Guidelines
- **Terminology:** Use precise and widely recognized academic or industry terminology for subjects, topics, and key concepts to maintain consistency and clarity.
- **Educational Level Selection:** If appropriate, restrict the educational level to one of the following: "Primary School", "Middle School", "High School", "College", "Graduate School", or "Competition" to ensure accurate categorization.

## Text
{{ text }}

## Output Format
<level>Educational Level</level>
<subject>Subject Area</subject>

<topic> Topics:
1. topic 1
2. topic 2
</topic>

<key_concept>
Key Concepts:
1. topic 1:
    1.1. key concept
    1.2. key concept
    ...
2. topic 2:
    2.1. key concept
    ... ...
</key_concept>

## Output

Figure 10: Prompt for extracting topics and key concepts from the document. The {{ text }} is the placeholders for inserting reference document.

# E  Examples about Extracted Topics and Key Concepts from Reference Document

In this section, we demonstrate some examples about extracted topics and key concepts from reference document by using previous prompt shown in Figure 10. We adopt a relative large Language model, Mistral-Large-Instruct-2407. The examples are shown in Figure 11 and 12. We can observe that Mistral-Large-Instruct-2407 is capable of extracting well-structured topics and their corresponding key concepts from the document.

```
<level>High School</level>
<subject>Trigonometry</subject>

<topic>
Topics:
1. Trigonometric Functions and Identities
2. Geometry on a Sphere
3. Applications of Trigonometry
4. Complex Numbers and Trigonometry
5. Derivations and Proofs in Trigonometry
</topic>

<key_concept>
Key Concepts:
1. Trigonometric Functions and Identities:
        1.1. Sine, Cosine, and Tangent Functions
        1.2. Trigonometric Identities (e.g., Pythagorean, Co-function, Sum and Difference)
        1.3. Law of Sines and Law of Cosines
        1.4. Inverse Trigonometric Functions
        1.5. Trigonometric Equations and Their Solutions

2. Geometry on a Sphere:
        2.1. Latitude and Longitude
        2.2. Circumference of a Circle on a Sphere
        2.3. Equation of a Circle on a Sphere
        2.4. Spherical Trigonometry
        2.5. Haversine Formula for Distance Calculation

3. Applications of Trigonometry:
        3.1. Circular Motion and Angular Velocity
        3.2. Calculating Angles in Right Triangles
        3.3. Real-world Problems Involving Trigonometry (e.g., Crossing a Canyon)
        3.4. Computing Sine and Cosine without a Calculator
        3.5. Practical Uses of Trigonometric Functions in Engineering and Physics

4. Complex Numbers and Trigonometry:
        4.1. Euler's Formula and De Moivre's Theorem
        4.2. Complex Numbers and Trigonometric Functions
        4.3. Roots of Unity and Their Relationship to Trigonometry
        4.4. Inverse Hyperbolic Functions (e.g., Arcsinh)
        4.5. Complex Numbers in Trigonometric Identities

5. Derivations and Proofs in Trigonometry:
        5.1. Derivation of Trigonometric Identities
        5.2. Proof of the Law of Sines and Law of Cosines
        5.3. Derivation of Sum and Difference Formulas
        5.4. Proof of the Haversine Formula
        5.5. Derivation of Heron's Formula
</key_concept>
```

Figure 11: An example about High School's Trigonometry.

<level>College</level>
<subject>Vector Calculus</subject>

<topic>
Topics:
1. Surface Integrals of Vector Fields
2. Oriented Surfaces
3. Unit Normal Vectors
4. Flux of a Vector Field
5. Parametric Surfaces
</topic>

<key_concept>
Key Concepts:
1. Surface Integrals of Vector Fields:
 1.1. Definition of surface integral of a vector field
 1.2. Flux of a vector field across a surface
 1.3. Application of surface integrals in fluid dynamics
 1.4. Evaluation of surface integrals using parametric surfaces
 1.5. Surface integrals over closed surfaces

2. Oriented Surfaces:
 2.1. Definition of an oriented surface
 2.2. Positive and negative orientations
 2.3. Unit normal vectors and their role in orientation
 2.4. Orientation conventions for closed surfaces
 2.5. Impact of orientation on surface integrals

3. Unit Normal Vectors:
 3.1. Definition and calculation of unit normal vectors
 3.2. Gradient vector and its role in finding normal vectors
 3.3. Normal vectors for surfaces given by z = f(x, y)
 3.4. Normal vectors for parametric surfaces
 3.5. Adjusting normal vectors to match desired orientation

4. Flux of a Vector Field:
 4.1. Definition of flux
 4.2. Physical interpretation of flux in fluid dynamics
 4.3. Calculation of flux using surface integrals
 4.4. Flux across closed surfaces
 4.5. Application of flux in Gauss's Law

5. Parametric Surfaces:
 5.1. Definition and representation of parametric surfaces
 5.2. Calculation of normal vectors for parametric surfaces
 5.3. Evaluation of surface integrals using parametric surfaces
 5.4. Parameterization of common surfaces (e.g., spheres, cylinders)
 5.5. Conversion between parametric and non-parametric forms
</key_concept>

Figure 12: An example about the College's Vector Calculus.

