# OpenReview forum: "Scaling Laws of Synthetic Data for Language Model"
_colmweb.org/COLM/2025/Conference — COLM 2025_

### Official Review · Reviewer_oLip · 2025-05-09

**Rating:** 7
**Confidence:** 3
**Ethics Flag:** 1

**Summary:**

This paper studies scaling laws for synthetic data for LLMs, with a specific focus on the mathematics domain. The authors propose SynthLLM which is a scalable framework that can transform pre-training corpora in high-quality synthetic datasets. The paper is well-written, and especially the discussion of scaling laws for synthetic data and the technical description of the SynthLLM approach are strong and in-depth. Scaling laws have received a lot of attention, but less so for synthetic data. This makes for a potentially significant contribution.

**Questions To Authors:**

Thanks for a very interesting paper! I have a few questions, and some suggestions:

Questions
- why did you choose for this particular domain, and how generalisable do you think the approach is?
- can you say bit more about the generated questions and answers. Did you (anecdotally) encounter mathematical mistakes?
- what do you think are the limits of the "predictable performance gains" you conjecture in section 5?
- can you say a bit more about the prompting strategy?
- The examples in Figure 11 and 12 do look a lot like typical mathematics syllabi -- does your approach work well on this domain since there are many such examples around?

Suggestions
- it would be good to be more explicit about the mathematics domain in the title and the first paragraphs of the introduction
- there were some mistakes in the bibliography (Muennighoff et al. and Toshniwal et al. both are listed twice)

**Reasons To Accept:**

- interesting topic (scaling laws for synthetic data)
- solid technical work
- detailed (automatic) evaluation of the new model, compared with a broad range of other models, and on a range of math benchmarks, with good results

**Reasons To Reject:**

- evaluation could have been more detailed: it would have been interesting to learn more about the prompting strategy (how were prompts designed? why only one prompt?), and to get more qualitative insight in the results (why does the proposed model fare better than other models? in which area's is it still lacking)
- no insight in the quality/correctness of the generated answers
- tied to a particular domain (mathematics) -- does the method only work for this specific and structured domain?

---

> ### Author Response · Authors · 2025-06-03
> **Author Response [1/3]**
>
> We would like to start by expressing our thanks to you for taking the time and effort to review our work, and for your recognition of our contributions.
>
> ------
> ### Weakness 1:
> Thank you for your valuable comments. We would like to address your questions regarding our prompting strategy and the qualitative insights into our results as follows:
> 1. Prompting Strategy:
>     1.1. Prompt Design Rationale and Objectives: Our motivation was to effectively leverage limited pre-training documents to scale high-quality questions with significant diversity (mentioned in Line 211-219). Therefore, we provided very detailed descriptions about how to induce diversity in question generation via our newly proposed Level-2 and Level-3 method. As detailed in Section 4.2 of our paper (and further illustrated in Appendix Figures 8 and 9), our prompt design for Level-2 and Level-3 follows a "concept extraction - random combination - reference-based generation" pipeline.
>     1.2. Regarding the Number of Prompts and Tuning (Why only one prompt? / Further tuning): We currently employ one core, structured prompt template for Level-2 and Level-3. We did not explore extensive prompt variations, as preliminary experiments showed that the current prompts already enabled the LLM (Mistral-Large-Instruct-2407) to accurately perform concept combinations and generate questions consistent with the reference content. Given the LLM’s strong instruction-following ability, we focused more on validating the effectiveness of the concept combination strategy itself in scaling data diversity and improving downstream model performance.
>
> 2. We attribute the superior performance of models trained with SYNTHLLM-generated data (from our Level-2 and Level-3 methods) primarily to the systematic enhancement of data diversity:
>     2.1. Enhanced Diversity: As shown in the ablation studies in Section 4.3 of our paper, by decomposing and recombining concepts, Level-2 and Level-3 can generate significantly more diverse questions than direct extraction baseline (Level-1). The histogram of question similarity in Figure 4 visually demonstrates that questions generated by Level-2 exhibit lower internal similarity (i.e., higher diversity) compared to Level-1.
>     2.2.  Better performance: The results in Table 3 from our ablation studies show that models trained on data generated by Level-2 and Level-3 consistently outperform models trained on Level-1 data across various benchmarks. These results directly verify the effectiveness of our methodical design.
>
> ------
>
> ### Weakness 2:
>
> Thanks for your comments. To clarify, we briefly outline our data quality control procedures and present our earlier preliminary experimental results evaluating the impact of filtering potentially noisy answers on model performance.
>
> 1. Our quality control procedures:
> In addition to the de-duplication and decontamination against evaluation benchmarks (mentioned in Sec.B), we also performed basic quality checks on the synthetic data, including checks for question formatting, answer format filtering (e.g., \boxed{}).
> 2. Limited Impact of Answer Quality Filtering on Performance:
> The decision not to conduct further answer verification was motivated by our prior preliminary experiments, which showed that removing noisy samples during SFT yielded only marginal gains in model performance. We describe these experiments below.
> Our preliminary experiments followed the same experimental setup as the ablation studies described in Section 4.3. We randomly selected 2,000 reference documents and generated 50 questions per document using our Level-2 method. For each question, we generated 8 candidate answers. For each question, we used Llama-3.1-70B-Instruct as an answer evaluator to score its 8 candidate answers and discarded any judged “potentially incorrect.” We then randomly picked one answer from the remaining candidates. Without filtering, one answer was randomly selected from all 8 candidates. In both cases, only one answer per question was retained for comparison.
> We then trained the Llama-3.1-8B model on these generated datasets. The evaluation results are shown in the table below. We evaluate each trained model on the MATH benchmark and compute the average accuracy across all other benchmarks. As shown, filtering out noisy samples results in only marginal gains in model performance.
> |ACC(%)|MATH|average|
> |--|--|--|
> |wo filtering|42.0|36.5|
> |w filtering|42.2|36.6|
>
> ------

---

> > ### Author Response · Authors · 2025-06-03
> > **Author Response [2/3]**
> >
> > ### Continue to Weakness 2:
> > 3. Similar Observations from Recent Work:
> > Recent studies [1,2,3] report consistent findings. As shown in Table 7 of [1], adding a validation step for synthetic data does not improve performance on mathematical reasoning tasks. Table 2 of [2] and Figure 6 of [3] further show that models trained even on entirely incorrect answers achieve performance comparable to those trained on correct answers. These results, along with our experiments, suggest that for SFT, model performance remains relatively robust to a certain degree of noise in answer correctness.
> > We believe that the primary role of SFT is to align models with target distributions and to activate their inherent reasoning abilities and knowledge acquired during pre-training rather than to enforce the verbatim correctness of every individual sample. Therefore, we prioritize expanding the scale and diversity of synthetic questions at this stage to better investigate the scaling laws of synthetic data.
> >
> >
> > We appreciate your thoughtful comments. We will incorporate this discussion more clearly in the revised version to better explain our design choices and motivations.
> >
> > -----
> >
> > ### Weakness 3:
> > Thank you for your comments. To explore the generalizability and effectiveness of our method, we conduct preliminary experiments in the coding domain.
> >
> > We adopt the same SYNTHLLM framework (described in Section 3) and follow the experimental setup outlined in Section 4.1. Specifically, we generate 2.2M samples using our Level-1 (direct extraction-based) baseline method, and 2.4M and 2.5M samples for our newly proposed Level-2 (single-document concept recombination) and Level-3 (multi-document graph-based concept recombination) methods, respectively. Mistral-Large-Instruct-2407 is used as the question generator, while Qwen2.5-Coder-32B-Instruct serves as the answer generator. We then train the Llama-3.1-8B base model on these datasets.
> > The evaluation results on HumanEval and MBPP are summarized in the below table. For comparison, we also include the performance of the instruction-tuned model (Llama-3.1-8B-Instruct) and a code-specific model (Deepseek-Coder-v1.5-Instruct).
> > |ACC(%)|Human|MBPP|
> > |--|--|--|
> > |Llama-3.1-8B-Instruct|72.6|71.2|
> > |Deepseek-Coder-v1.5-Instruct|75.6|73.4|
> > |SYNTHLLM Level-1 |73.2|67.3|
> > |SYNTHLLM Level-123 |78.7|72.5|
> >
> > SYNTHLLM Level-1 represents the model trained solely on Level-1 data, while the Level-123 model denotes the model trained on the entire synthetic dataset (combined data from 3 methods).
> > Our newly proposed Level-2 and Level-3 methods yield significant improvements in coding performance over the Level-1 baseline. SYNTHLLM Level-123 outperforms the instruction-tuned model and achieves performance comparable to the code-specific model, demonstrating the generalizability and effectiveness of our approach.
> >
> > -------
> > ### Question 1:
> > We chose the mathematical reasoning domain for its well-established evaluation protocols and metrics, essential for analyzing scaling behavior (as mentioned in Line 121-124). Moreover, complex reasoning (e.g., math reasoning) is a core capability for LLMs, and the domain’s structured knowledge and logical rigor provide an ideal testbed to evaluate the quality and effectiveness of data generated by SYNTHLLM.
> >
> > The above results on the coding domain have demonstrated the effectiveness and generalizability of our approach.
> >
> > -----
> >
> > ### Question 2:
> >
> > Thanks for your questions.
> > For “generated questions and answers”:
> > We have added more details about the prompting process for question generation and expanded the discussion on answer correctness verification. Additional information about our generated dataset is also provided in Section B (“Our Dataset” part). Thank you for your valuable suggestion; we will ensure these clarifications are clearly reflected in the revised version.
> >
> > For “encounter mathematical mistakes”:
> > Yes, in our preliminary experiments on answer verification, we employed Llama-3.1-70B-Instruct as the evaluator and observed that some answers exhibited computational errors during the reasoning process. As mentioned, both our preliminary experiments and recent work [1,2,3] show that SFT model performance remains robust despite a certain degree of noise in answer correctness.
> >
> > ------

---

> > > ### Author Response · Authors · 2025-06-03
> > > **Author Response [3/3]**
> > >
> > > ### Question 3:
> > > Thank you for raising this insightful question. The predictable performance gains discussed in Section 5 are estimated based on our rectified power law. It is worth noting that, as the scale of the synthetic dataset increases, the marginal performance gains gradually diminish, reflecting a characteristic property of the power law. As illustrated in Table 1, the magnitude of performance improvement decreases progressively with larger dataset scales.
> > >
> > >
> > > Hence, as the dataset grows, further innovations in training methods may be needed to improve the efficiency of data utilization. We see this as an important direction for future work.
> > >
> > > ----
> > >
> > > ### Question 4:
> > > Thanks for your question. The details about prompting strategy have been mentioned in the first response. We will ensure that this discussion is made clearer in the revised version of our paper.
> > >
> > > ------
> > >
> > > ### Question 5:
> > >
> > > We thank the reviewer for raising this insightful observation.
> > >
> > > First, we note that not only mathematics but also domains such as coding, physics, law, and medicine typically have well-structured curricula and systematic syllabi. Additionally, abundant web documents [4] provide extensive resources covering these domains. Therefore, our proposed approach can naturally generalize and extend to these diverse domains. The preliminary experimental results on the coding domain have demonstrated the effectiveness and generalizability of our approach.
> > >
> > > Second, our method does not simply replicate existing questions or syllabi. Instead, our Level-2 and Level-3 methods systematically sample and combine knowledge concepts in a more diversified manner, thereby enabling more scalable and diverse synthetic question generation.
> > >
> > > Finally, as demonstrated in the ablation studies presented in Section 4.3, even under controlled settings with limited document resources, our Level-2 and Level-3 methods consistently outperform existing data augmentation methods in terms of data expansion efficiency and downstream model performance. This further highlights the intrinsic effectiveness and advantages of our proposed methodology.
> > >
> > > ------
> > > ### Suggestions
> > >
> > > Thanks for your suggestions. We will carefully review and revise these points in the updated version.
> > >
> > > --------
> > > [1]. MathScale: Scaling Instruction Tuning for Mathematical Reasoning, ICML 2024.
> > > [2]. LLMs Can Easily Learn to Reason from Demonstrations Structure, not content, is what matters!, arxiv 2025.
> > > [3]. Cognitive Behaviors that Enable Self-Improving Reasoners, or, Four Habits of Highly Effective STaRs, arxiv 2025.
> > > [4]. The FineWeb Datasets: Decanting the Web for the Finest Text Data at Scale, NeurIPS 2024.

---

> > > ### Comment · Reviewer_oLip · 2025-06-03
> > >
> > > Thank you for this additional information. I remain positive about this work, and I feel my original scores reflect this well.

---

> > > > ### Author Response · Authors · 2025-06-04
> > > > **Response**
> > > >
> > > > Dear Reviewer oLip,
> > > >
> > > > Thanks for your recognition for our work!
> > > > We sincerely appreciate your invaluable feedback throughout the review process and will incorporate your suggestions as we revise the paper.
> > > >
> > > > The Authors.

---

### Official Review · Reviewer_Pj9b · 2025-05-10

**Rating:** 6
**Confidence:** 3
**Ethics Flag:** 1

**Summary:**

The paper works on a new synthetic data generation approach. As the main contributions, they highlight their pipeline for synthetic data generation that achieves better performance on mathematical benchmarks than the previous methods, as well as the fact that their mehod follows the rectified scaling law. Their experiments also showcase that smaller models need more synthetic tokens to achieve optimal performance in comparison to the larger counterparts. Final conclusion they make is that synthetic data is a reliable alternative to raw pre-training data.

**Questions To Authors:**

1. When evaluating the predictive accuracy of the scaling curves, why is model with 1B parameters excluded? You have mentioned that the model benefits less from synthetic data increase, does it have anything to do with this? In addition, have you tried to understand why this happens?
2. When creating the cold-start domain classifier, just to confirm - random sampling of negative examples is done assuming everything from Fineweb-Edu is a negative sample? I am not sure I understood this correctly
3. Do you have some examples of the output of concept combination sampling?
4. In Sec. 3.3, you mention that you don’t incoroporate verification as preliminary experiments indicate that your synthetic data is of satisfactory quality. What are those preliminary experiments? Does this refer to the Table 2? If that is the case, I don’t think this is rigorous enough, especially given the concerns I mentioned in ‘Reasons to reject’ above
5. Not urgent, but it would be nice to have some short description of baselines, maybe in appendix
6. In Sec 4.2, line 315, you mention that you train models sing full synthetic dataset and a subset of 3.2M samples. How were these samples chosen?
7. In ablation study where you compare the method with augmentation methods on limited reference documents, I’m a little confused with the methodology. Why does the augmentation start from 6500 seed questions?

**Reasons To Accept:**

The paper definitely tackles one of the common problems when training language models that does not seem to have a straightforward solution - quality data scarcity. The ideas on how to generate quality synthetic data overall seem sound. I also appreciate the effort in ablation studies to isolate the effect of each part of their pipeline.

**Reasons To Reject:**

In my opinion, there are two big issues with the current approach. The first one is that the synthetic data generation pipeline was tested only on one type of tasks - maths questions. For the claims that paper makes to be valid, I believe this should have been tested over more domains. I understand that this might be time-intensive, but at least adding a couple more types of tasks is necessary. Second, the dataset that was used to create synthetic data was based on CommonCrawl. Given that there was no targeted evaluation on contamination of this dataset with the content of datasets the authors evaluate on, I cannot be sure if the results are actually higher due to that, or superiority of the approach. To be fair, this is also true for some (if not all) of the baselines they compare to, so it is hard to pinpoint generally where the gains came from.

Typos:
Typo in line 200 - Let Let

More in the questions below.

---

> ### Author Response · Authors · 2025-06-03
> **Author Response [1/3]**
>
> We would like to start by expressing our thanks to you for taking the time and effort to review our work, and for your recognition of our contributions.
>
> ------
> ### Weakness 1:
> Thank you for your comments. To explore the generalizability and effectiveness of our method, we conduct preliminary experiments in the coding domain.
> We adopt the same SYNTHLLM framework (described in Section 3) and follow the experimental setup outlined in Section 4.1. Specifically, we generate 2.2M samples using our Level-1 (direct extraction-based) baseline method, and 2.4M and 2.5M samples for our newly proposed Level-2 (single-document concept recombination) and Level-3 (multi-document graph-based concept recombination) methods, respectively. Mistral-Large-Instruct-2407 is used as the question generator, while Qwen2.5-Coder-32B-Instruct serves as the answer generator. We then train the Llama-3.1-8B base model on these datasets.
> The evaluation results on HumanEval and MBPP are summarized in the below table. For comparison, we also include the performance of the instruction-tuned model (Llama-3.1-8B-Instruct) and a code-specific model (Deepseek-Coder-v1.5-Instruct).
>
> |ACC(%)|Human|MBPP|
> |--|--|--|
> |Llama-3.1-8B-Instruct|72.6|71.2|
> |Deepseek-Coder-v1.5-Instruct|75.6|73.4|
> |SYNTHLLM Level-1 |73.2|67.3|
> |SYNTHLLM Level-123 |78.7|72.5|
>
> SYNTHLLM Level-1 represents the model trained solely on Level-1 data, while the Level-123 model denotes the model trained on the entire synthetic dataset (combined data from 3 methods).
> Our newly proposed Level-2 and Level-3 methods yield significant improvements in coding performance over the Level-1 baseline. SYNTHLLM Level-123 outperforms the instruction-tuned model and achieves performance comparable to the code-specific model, demonstrating the generalizability and effectiveness of our approach.
>
> -----
>
> ### Weakness 2:
> Thanks for your comments. In fact, we performed a thorough decontamination process, as described in Lines 601–605 of Appendix. Following the protocol in [1], we conducted detailed string-matching de-duplication of questions against each evaluation benchmark to ensure thorough decontamination. We appreciate your feedback and will make sure to present these details more clearly in main text.
>
> ----
> ### Question 1:
> Thanks for your questions.
> For 1B predictive accuracy:
> 1. Task Difficulty and Model Capability: The MATH benchmark is particularly challenging for smaller models. As shown in Table 2, with the current size of the synthetic dataset, the 1B model achieves a peak accuracy of only 37–38%. At this relatively low performance level, which is significantly distant from saturation for this complex task, the learning curves are more unstable and prone to fluctuations, making it difficult to obtain a reliable scaling trend with current data size.
> 2. Impact of Pre-training knowledge: Larger models, such as the 3B and 8B variants, benefit from a more substantial amount of pre-learned knowledge from pre-training relevant to complex math reasoning tasks. This corresponds to a larger $D_l$ in the Rectified Scaling Law (Figure 1), providing a more stable baseline for fitting scaling curves and yielding behavior more consistent with the expected empirical scaling trends (as shown in Figure 1 and discussed in Sec. 2.2).
>
> 3. We believe that consistent trends in the 3B and 8B models indicate that synthetic data is a scalable resource yielding predictable performance gains.
>
>
> For “model benefits less from synthetic data increase”:
> We believe the observed behavior of the 1B model may be attributed to the following factors:
> 1. Model capacity limitations are the primary cause. The 1B model, due to its relatively small parameter count, has constrained representational power. This constrains its ability to efficiently fit and generalize patterns from complex reasoning tasks. As more synthetic data is added, the model continues to improve, but the marginal gains diminish more rapidly compared to larger models. This is reflected in the smaller decay exponent observed in Figure 1. The consistent observations are also shown in paper of scaling law on real data [2] (Figure 5 and 7).
> 2. Regarding pre-training knowledge, although all models possess prior knowledge from pre-training, larger models (such as the 3B and 8B variants) generally encode more and broader knowledge that is more relevant to complex reasoning tasks. As a result, larger models can better leverage synthetic data for knowledge transfer and generalization, whereas the 1B model benefits less from the same amount of additional data. This is reflected in the smaller $D_l$ value observed in the fitted scaling law for the 1B model. Similar observations have also been discussed in Figure 1 of the OpenAI paper on the scaling laws of transfer learning [2] and in Figure 3 of the rectified scaling law study on real data [3].
>
> We will incorporate these explanations more explicitly in the revised version.

---

> > ### Author Response · Authors · 2025-06-03
> > **Author Response [2/3]**
> >
> > ### Question 2:
> >
> > Thanks for your question.
> > Your understanding is correct. Following [1], in constructing the cold-start domain classifier, we obtain negative samples by randomly sampling from the Fineweb-Edu corpus.
> >
> > Due to the lack of explicitly labeled negatives at this stage, we use random sampling to approximate a non-target domain distribution, which is inherently different from our specific target domain (e.g., mathematics). Given the size and diversity of Fineweb-Edu, randomly selected documents are unlikely to belong to any specific, narrow domain.
> >
> > The cold-start classifier serves to perform large-scale initial filtering, identifying a smaller candidate set ($D^0$) more likely to be domain-relevant. This set is then used to iteratively train a fine-grained domain classifier (as shown in Figure 3). In later stages, we refine the document pool using more precise labeling techniques, such as LLM scoring, to ensure high-quality references for synthetic data generation.
> >
> > ----
> > ### Question 3:
> > Yes. We provide several examples of the output from concept combination sampling (Level-2 method). The “Selected Concepts” part denotes the concept combination and the “Question” part denotes the generated question based on this concept combination.
> >
> > Q1:
> > Selected Concepts:
> > - Triangle Classifications: Isosceles Triangle
> > - Median of a Triangle: Median Formula for Isosceles Triangle
> > - Properties of Medians: Equal Area Division by Each Median
> >
> > Question:
> > In an isosceles triangle \( \triangle ABC \), the sides \( AB \) and \( AC \) are each 10 units long, and the base \( BC \) is 6 units long. Calculate the length of the median from vertex \( A \) to the midpoint of \( BC \). Additionally, prove that this median divides the triangle into two smaller triangles of equal area.
> >
> >
> > Q2:
> > Selected Concepts:
> > - Inscribed Angle Theorem
> > - Alternate Segment Theorem
> > - Cyclic Quadrilateral Theorem
> >
> > Question:
> > In circle \( O \), let \( A \) and \( B \) be points on the circle such that \( AB \) is not a diameter. A tangent at point \( A \) intersects the extension of \( OB \) at point \( T \). Let \( C \) be a point on the circle such that \( \angle ACB = 30^\circ \). Determine the measure of \( \angle TAC \) and prove that quadrilateral \( AOTC \) is cyclic.
> >
> > Q3:
> > Selected Concepts:
> > - Calculus: Limits
> > - Calculus: Derivatives
> > - Linear Algebra: Vectors
> >
> > Question:
> > Suppose $\mathbf{v}(t) = (t^2, e^{-t}, \sin(t))$ is a vector-valued function. Calculate the limit $\lim_{t \to 0} \frac{\mathbf{v}(t) - \mathbf{v}(0)}{t}$ and determine the derivative of $\mathbf{v}(t)$ at $t = 0$.
> >
> > -----
> >
> > [1]. MAmmoTH2: Scaling Instructions from the Web, NeurIPS 2024.
> > [2]. Scaling Laws for Transfer, arxiv 2021.
> > [3]. Selecting Large Language Model to Fine-tune via Rectified Scaling Law, ICML 2024

---

> > > ### Author Response · Authors · 2025-06-03
> > > **Author Response [3/3]**
> > >
> > > ### Question 4:
> > >
> > > Thanks for your comments.
> > >
> > > 1. Limited Impact of Answer Quality Filtering on Performance:
> > > The decision not to conduct further answer verification was motivated by our prior preliminary experiments, which showed that removing noisy samples during SFT yielded only marginal gains in model performance. We describe these experiments below.
> > > Our preliminary experiments followed the same experimental setup as the ablation studies described in Section 4.3. We randomly selected 2,000 reference documents and generated 50 questions per document using our Level-2 method. For each question, we generated 8 candidate answers. For each question, we used Llama-3.1-70B-Instruct as an answer evaluator to score its 8 candidate answers and discarded any judged “potentially incorrect.” We then randomly picked one answer from the remaining candidates. Without filtering, one answer was randomly selected from all 8 candidates. In both cases, only one answer per question was retained for comparison.
> > > We then trained the Llama-3.1-8B model on these generated datasets. The evaluation results are shown in the table below. We evaluate each trained model on the MATH benchmark and compute the average accuracy across all other benchmarks.  As shown, filtering out noisy samples results in only marginal gains in model performance.
> > > |ACC(%)|MATH|average|
> > > |--|--|--|
> > > |wo filtering|42.0|36.5|
> > > |w filtering|42.2|36.6|
> > >
> > > 2. Similar Observations from Recent Work:
> > > Recent studies [4,5,6] report consistent findings. As shown in Table 7 of [4], adding a validation step for synthetic data does not improve performance on mathematical reasoning tasks. Table 2 of [5] and Figure 6 of [6] further show that models trained even on entirely incorrect answers achieve performance comparable to those trained on correct answers. These results, along with our experiments, suggest that for SFT, model performance remains relatively robust to a certain degree of noise in answer correctness.
> > > We believe that the primary role of SFT is to align models with target distributions and to activate their inherent reasoning abilities and knowledge acquired during pre-training rather than to enforce the verbatim correctness of every individual sample. Therefore, we prioritize expanding the scale and diversity of synthetic questions at this stage to better investigate the scaling laws of synthetic data.
> > > We appreciate your thoughtful comments. We will incorporate this discussion more clearly in the revised version to better explain our design choices and motivations.
> > >
> > > As mentioned above, we performed a thorough decontamination process, as described in Lines 601–605 of Appendix.
> > >
> > > ----
> > >
> > > ### Question 5:
> > >
> > > Thanks for your suggestions. We have provided short descriptions of baseline methods in Sec.B of Appendix (Line 617-624).
> > >
> > > ----
> > >
> > > ### Question 6:
> > > Sorry for the confusion. We uniformly sample 3.2M examples from the full dataset. We will clarify this point more explicitly in the revised version.
> > >
> > > -----
> > >
> > > ### Question 7:
> > >
> > > Sorry for the possible confusion. For rephrasing and persona augmentation, we need seed questions to conduct data augmentation. Therefore, we first adopt our Level-1 method to extract existing questions from the 2,000 fixed reference documents, which yields approximately 6,500 questions as the seed dataset.
> > > We appreciate your question and will make this explanation clearer in the revised version.
> > >
> > > -----
> > >
> > >
> > > [4]. MathScale: Scaling Instruction Tuning for Mathematical Reasoning, ICML 2024
> > > [5]. LLMs Can Easily Learn to Reason from Demonstrations Structure, not content, is what matters!, arxiv 2025.
> > > [6]. Cognitive Behaviors that Enable Self-Improving Reasoners, or, Four Habits of Highly Effective STaRs, arxiv 2025.

---

> > > > ### Comment · Reviewer_Pj9b · 2025-06-03
> > > > **Response**
> > > >
> > > > Thank you for clarifications!
> > > >
> > > > It's encouraging to see that there are indicators this might generalize to other (though still similar) domains, it is a good start. Thank you also for providing examples, and answering some of the questions I was confused about. I think the added analysis about quality of the synthetic data and the indicators about generalization improved my initial impression of the paper.

---

> > > > > ### Author Response · Authors · 2025-06-04
> > > > > **Response**
> > > > >
> > > > > Dear Reviewer Pj9b,
> > > > >
> > > > > Thank you very much for your thoughtful feedback. We are glad to hear that our responses and the additional analysis have helped clarify your concerns and improved your impression of the paper.
> > > > >
> > > > > If there are any remaining questions or concerns, we would be more than happy to further address them. We would also appreciate it if you would consider reflecting this updated assessment in your final score.
> > > > >
> > > > > Looking forward to your reply.
> > > > > Thank you again for your time and valuable feedback!
> > > > >
> > > > > The Authors.

---

> > > > > > ### Comment · Reviewer_Pj9b · 2025-06-06
> > > > > > **Response**
> > > > > >
> > > > > > It seems to me that the additional content you provided is heading in the right direction, so I increased the score. I don't have any additional questions, now it is more a matter of combining the new analyses into the paper (and potentially adding more experiments in different domains).

---

> > > > > > > ### Author Response · Authors · 2025-06-06
> > > > > > > **Response**
> > > > > > >
> > > > > > > Dear Reviewer Pj9b,
> > > > > > >
> > > > > > > Thanks for your recognition for our work! We are very glad that our response has addressed your concern. We sincerely appreciate your invaluable feedback throughout the review process and will incorporate your suggestions as we revise the paper.
> > > > > > >
> > > > > > > The Authors.

---

### Official Review · Reviewer_nd1f · 2025-05-17

**Rating:** 7
**Confidence:** 3
**Ethics Flag:** 1

**Summary:**

This paper investigates whether synthetic data can provide predictable scaling benefits for language models similar to those observed with natural pre-training data. The authors introduce SYNTHLLM, a framework that transforms pre-training corpora into synthetic datasets. SynthLLM filters Fineweb-Edu to get documents from the mathematical reasoning domain, then questions and answers are generated for these documents via prompting. Three different methods for question generation are explored. SynthLLM is evaluated across six mathematical reasoning benchmarks. The study also shows that synthetic data generated by SYNTHLLM follows a rectified scaling law across various model sizes.

**Questions To Authors:**

Typos:
- 319: "OpenMathInstruct-2, a high-quality math dataset based on MATH and GSM8K." (the phrase is not finished)
- 363: \cite{} for citations
- 378: approachs

**Reasons To Accept:**

- The paper clearly demonstrates the advantages of their synthetic data generation framework.
- Good ablation studies and comparison to other data augmentation methods are included.
- Very well-written and easy to follow

**Reasons To Reject:**

- No qualitative checks of generated synthetic data were conducted. Even if synthetic data showed gains in performance, it'd be interesting to analyse the quality level of the generated data.
- The study is limited to one domain: mathematical reasoning.

---

> ### Author Response · Authors · 2025-06-03
> **Author Response [1/2]**
>
> We would like to start by expressing our thanks to you for taking the time and effort to review our work, and for your recognition of our contributions.
>
> -----
>
> ### Weakness 1:
>
> Thanks for your comments. To clarify, we briefly outline our data quality control procedures and present our earlier preliminary experimental results evaluating the impact of filtering potentially noisy answers on model performance.
>
> 1. Our quality control procedures:
> In addition to the de-duplication and decontamination against evaluation benchmarks (mentioned in Sec.B), we also performed basic quality checks on the synthetic data, including checks for question formatting, answer format filtering (e.g., \boxed{}).
>
> 2. Limited Impact of Answer Quality Filtering on Performance:
> The decision not to conduct further answer verification was motivated by our prior preliminary experiments, which showed that removing noisy samples during SFT yielded only marginal gains in model performance. We describe these experiments below.
> Our preliminary experiments followed the same experimental setup as the ablation studies described in Section 4.3. We randomly selected 2,000 reference documents and generated 50 questions per document using our Level-2 method. For each question, we generated 8 candidate answers. For each question, we used Llama-3.1-70B-Instruct as an answer evaluator to score its 8 candidate answers and discarded any judged “potentially incorrect.” We then randomly picked one answer from the remaining candidates. Without filtering, one answer was randomly selected from all 8 candidates. In both cases, only one answer per question was retained for comparison. We then trained the Llama-3.1-8B model on these generated datasets. The evaluation results are shown in the table below. We evaluate each trained model on the MATH benchmark and compute the average accuracy across all other benchmarks.  As shown, filtering out noisy samples results in only marginal gains in model performance.
> |ACC(%)|MATH|average|
> |--|--|--|
> |wo filtering|42.0|36.5|
> |w filtering|42.2|36.6|
>
> 3. Similar Observations from Recent Work:
> Recent studies [1,2,3] report consistent findings. As shown in Table 7 of [1], adding a validation step for synthetic data does not improve performance on mathematical reasoning tasks. Table 2 of [2] and Figure 6 of [3] further show that models trained even on entirely incorrect answers achieve performance comparable to those trained on correct answers. These results, along with our experiments, suggest that for SFT, model performance remains relatively robust to a certain degree of noise in answer correctness.
> We believe that the primary role of SFT is to align models with target distributions and to activate their inherent reasoning abilities and knowledge acquired during pre-training rather than to enforce the verbatim correctness of every individual sample. Therefore, we prioritize expanding the scale and diversity of synthetic questions at this stage to better investigate the scaling laws of synthetic data.
> We appreciate your thoughtful comments. We will incorporate this discussion more clearly in the revised version to better explain our design choices and motivations.
> ----
> [1]. MathScale: Scaling Instruction Tuning for Mathematical Reasoning, ICML 2024
> [2]. LLMs Can Easily Learn to Reason from Demonstrations Structure, not content, is what matters!, arxiv 2025.
> [3]. Cognitive Behaviors that Enable Self-Improving Reasoners, or, Four Habits of Highly Effective STaRs, arxiv 2025.

---

> > ### Author Response · Authors · 2025-06-03
> > **Author Response [2/2]**
> >
> > ### Weakness 2:
> >
> > Thank you for your comments. To explore the generalizability and effectiveness of our method, we conduct preliminary experiments in the coding domain.
> >
> > We adopt the same SYNTHLLM framework (described in Section 3) and follow the experimental setup outlined in Section 4.1. Specifically, we generate 2.2M samples using our Level-1 (direct extraction-based) baseline method, and 2.4M and 2.5M samples for our newly proposed Level-2 (single-document concept recombination) and Level-3 (multi-document graph-based concept recombination) methods, respectively. Mistral-Large-Instruct-2407 is used as the question generator, while Qwen2.5-Coder-32B-Instruct serves as the answer generator. We then train the Llama-3.1-8B base model on these datasets.
> >
> > The evaluation results on HumanEval and MBPP are summarized in the below table. For comparison, we also include the performance of the instruction-tuned model (Llama-3.1-8B-Instruct) and a code-specific model (Deepseek-Coder-v1.5-Instruct).
> >
> > |ACC(%)|Human|MBPP|
> > |--|--|--|
> > |Llama-3.1-8B-Instruct|72.6|71.2|
> > |Deepseek-Coder-v1.5-Instruct|75.6|73.4|
> > |SYNTHLLM Level-1 |73.2|67.3|
> > |SYNTHLLM Level-123 |78.7|72.5|
> >
> >
> > SYNTHLLM Level-1 represents the model trained solely on Level-1 data, while the Level-123 model denotes the model trained on the entire synthetic dataset (combined data from 3 methods).
> > Our newly proposed Level-2 and Level-3 methods yield significant improvements in coding performance over the Level-1 baseline. SYNTHLLM Level-123 outperforms the instruction-tuned model and achieves performance comparable to the code-specific model, demonstrating the generalizability and effectiveness of our approach.
> >
> > ------
> >
> > ### Questions:
> > Thank you for pointing out these typos. We will correct them in the updated version.

---

> > > ### Comment · Reviewer_nd1f · 2025-06-05
> > >
> > > Thank you for your detailed response! It clarifies well potential weaknesses.

---

> > > > ### Author Response · Authors · 2025-06-06
> > > > **Response**
> > > >
> > > > Dear Reviewer nd1f,
> > > >
> > > > Thanks for your recognition for our work! We are very glad that our response has addressed your concern.
> > > > We sincerely appreciate your invaluable feedback throughout the review process and will incorporate your suggestions as we revise the paper.
> > > >
> > > > The Authors.

---

### Official Review · Reviewer_xJ12 · 2025-05-19

**Rating:** 6
**Confidence:** 3
**Ethics Flag:** 1

**Summary:**

This paper proposes SYNTHLLM, a scalable framework for generating high-quality synthetic datasets for LLM fine-tuning (for math, reasoning). The central contribution is an empirical demonstration that synthetic data can exhibit rectified scaling laws similar to organic pretraining data. The authors evaluate multiple methods for question generation, culminating in a graph-based Level-3 strategy that combines high-level concepts across documents. They show that SYNTHLLM outperforms existing synthetic datasets and augmentation strategies across several math benchmarks.

**Questions To Authors:**

Please check the previous section

**Reasons To Accept:**

* Proposing a high-quality synthetic data generation technique

* SYNTHLLM introduces a novel hierarchical and multi-level data synthesis framework, culminating in a graph-based sampling method that improves diversity and scalability.

* Evaluations span multiple model sizes (1B to 8B) and six math reasoning benchmarks, showing consistent and significant improvements over prior baselines.

* Scaling law for synthetic data for the first time: The demonstration that synthetic data adheres to the rectified scaling law is well-executed.

**Reasons To Reject:**

* The paper lacks any justification for why synthetic data should follow similar laws to real data.

* The paper focuses on scaling up the number of synthetic tokens but does not explore how quality variation in synthetic data affects the scaling curves.

* There’s no human or automatic metric evaluation of correctness or factual consistency in the generated Q&A pairs.

* All experiments are on math and reasoning. The paper claims SYNTHLLM can generalize to other domains (e.g., code, physics), but provides no evidence or preliminary experiments in such domains.

* The graph-based concept sampling in Level-3 is novel, but the paper provides little analysis of the graph structure (e.g., degree distribution, sparsity, connectedness). How robust is performance to graph quality?

* Many baselines (e.g., MAmmoTH2, JiuZhang3.0) use smaller training sizes than SYNTHLLM (e.g., 6M vs. 7.4M). A controlled size-matched comparison would be more informative.

* While answer generation is described, no rigorous filtering or correctness check is implemented. This is surprising, especially given the complexity of math problems.

---

> ### Author Response · Authors · 2025-06-03
> **Author Response [1/3]**
>
> We would like to start by expressing our thanks to you for taking the time and effort to review our work, and for your recognition of our contributions.
>
> ----
> ### Weakness 1:
> Thank you for your comments. Although our work primarily focuses on the empirical demonstration that synthetic data follows rectified scaling laws, the underlying rationale can be explained from several perspectives:
> 1. **Origin in Real Data**: Our SYNTHLLM framework generates synthetic data deeply grounded in real pre-training corpora. It systematically extracts and recombines knowledge concepts from these documents, while also using documents as references to generate new questions (Section 3). This inherent linkage ensures that the synthetic data preserves the essential knowledge structures and statistical properties of organic data, which are critical for effective model learning.
>
> 2. **Applicability of the Rectified Scaling Law**: A key finding of our study is that our synthetic data consistently adheres to the Rectified Scaling Law (as shown in Eq. 2 and Figure 1). This law explicitly accounts for the pre-learned knowledge $D_l$ (shown in Eq.2)​ that the model has acquired during its initial pre-training on large-scale real-world datasets. As a result, the scaling behavior observed on synthetic data closely aligns with that observed during fine-tuning on real data.
>
> 3. **Consistency in Learning Objectives**: Regardless of data source, LLMs are trained to optimize similar learning objectives, such as capturing conceptual relationships and reasoning patterns. Our SYNTHLLM framework is designed to generate synthetic data that preserves those key characteristics of real data crucial for model learning, enabling the scaling behavior during training to closely align with that seen on real-world datasets.
>
> Thanks again for your question. We will add these discussions in our revised version.
>
> ------
>
> ### Weakness 2 and Weakness 3:
> Thanks for your comments. To clarify, we briefly outline our data quality control procedures and present our earlier preliminary experimental results evaluating the impact of filtering potentially noisy answers on model performance.
>
> 1. **Our quality control procedures:**
> In addition to the de-duplication and decontamination against evaluation benchmarks (mentioned in Sec.B), we also performed basic quality checks on the synthetic data, including checks for question formatting, answer format filtering (e.g., \boxed{}).
>
> 2. **Limited Impact of Answer Quality Filtering on Performance:**
> The decision not to conduct further answer verification was motivated by our prior preliminary experiments, which showed that removing noisy samples during SFT yielded only marginal gains in model performance. We describe these experiments below.
> Our preliminary experiments followed the same experimental setup as the ablation studies described in Section 4.3. We randomly selected 2,000 reference documents and generated 50 questions per document using our Level-2 method. For each question, we generated 8 candidate answers. For each question, we used Llama-3.1-70B-Instruct as an answer evaluator to score its 8 candidate answers and discarded any judged “potentially incorrect.” We then randomly picked one answer from the remaining candidates. Without filtering, one answer was randomly selected from all 8 candidates. In both cases, only one answer per question was retained for comparison.
> We then trained the Llama-3.1-8B model on these generated datasets. We evaluate each trained model on the MATH benchmark and compute the average accuracy across all other benchmarks. As shown the below table, filtering out noisy samples results in only marginal gains in model performance.
> |ACC(%)|MATH|average|
> |--|--|--|
> |wo filtering|42.0|36.5|
> |w filtering|42.2|36.6|

---

> > ### Author Response · Authors · 2025-06-03
> > **Author Response [2/3]**
> >
> > ### Continue to Weakness 2 and 3:
> > 3.  **Similar Observations from Recent Work:**
> > Recent studies [1,2,3] report consistent findings. As shown in Table 7 of [1], adding a validation step for synthetic data does not improve performance on mathematical reasoning tasks. Table 2 of [2] and Figure 6 of [3] further show that models trained even on entirely incorrect answers achieve performance comparable to those trained on correct answers. These results, along with our experiments, suggest that for SFT, model performance remains relatively robust to a certain degree of noise in answer correctness.
> >
> > Therefore, we prioritize expanding the scale and diversity of synthetic questions at this stage to better investigate the scaling laws of synthetic data.
> > We appreciate your thoughtful comments. We will incorporate this discussion more clearly in the revised version to better explain our design choices and motivations.
> >
> > [1]. MathScale: Scaling Instruction Tuning for Mathematical Reasoning, ICML 2024
> > [2]. LLMs Can Easily Learn to Reason from Demonstrations Structure, not content, is what matters!, arxiv 2025.
> > [3]. Cognitive Behaviors that Enable Self-Improving Reasoners, or, Four Habits of Highly Effective STaRs, arxiv 2025.
> >
> > -----
> >
> > ### Weakness 7:
> > Thank you for your comments. Below, we provide our response. We believe that the primary role of SFT is to align models with target distributions and to activate their inherent reasoning abilities and knowledge acquired during pre-training.
> > Recent studies [2,3] have shown that for SFT in math reasoning tasks, the proper reasoning process and structure are more critical than the literal accuracy of each local step or final answer. Even when the training data contains incorrect answers, models are still able to learn effective reasoning patterns and achieve performance comparable to models trained on entirely correct data (as shown in Table 2 of [2] and Figure 6 of [3]). Additionally, [1] argues that the main goal of SFT is to align models with the task distribution rather than to enforce the verbatim correctness of every individual sample.
> >
> > We thank the reviewer for raising this important point. We will explicitly incorporate this discussion in the revised version.
> > While we have not performed instance-level answer verification in this work, we recognize its importance and the significant potential. Testing-time inference methods may help obtain higher-quality answers and further improve the efficiency of scaling synthetic data. We will continue to explore this important direction.
> >
> > -----
> >
> > ### Weakness 4:
> >
> > Thank you for your comments. To explore the generalizability and effectiveness of our method, we conduct preliminary experiments in the coding domain.
> > We adopt the same SYNTHLLM framework (described in Section 3) and follow the experimental setup outlined in Section 4.1. Specifically, we generate 2.2M samples using our Level-1 (direct extraction-based) baseline method, and 2.4M and 2.5M samples for our newly proposed Level-2 (single-document concept recombination) and Level-3 (multi-document graph-based concept recombination) methods, respectively. Mistral-Large-Instruct-2407 is used as the question generator, while Qwen2.5-Coder-32B-Instruct serves as the answer generator. We then train the Llama-3.1-8B base model on these datasets.
> > The evaluation results on HumanEval and MBPP are summarized in the below table. For comparison, we also include the performance of the instruction-tuned model (Llama-3.1-8B-Instruct) and a code-specific model (Deepseek-Coder-v1.5-Instruct).
> > |ACC(%)|Human|MBPP|
> > |--|--|--|
> > |Llama-3.1-8B-Instruct|72.6|71.2|
> > |Deepseek-Coder-v1.5-Instruct|75.6|73.4|
> > |SYNTHLLM Level-1 |73.2|67.3|
> > |SYNTHLLM Level-123 |78.7|72.5|
> >
> > SYNTHLLM Level-1 represents the model trained solely on Level-1 data, while the Level-123 model denotes the model trained on the entire synthetic dataset (combined data from 3 methods).
> > Our newly proposed Level-2 and Level-3 methods yield significant improvements in coding performance over the Level-1 baseline. SYNTHLLM Level-123 outperforms the instruction-tuned model and achieves performance comparable to the code-specific model, demonstrating the generalizability and effectiveness of our approach.

---

> > ### Author Response · Authors · 2025-06-03
> > **Author Response [3/3]**
> >
> > ### Weakness 5:
> > Thank you for your questions regarding the graph structure analysis.
> > 1. As described in Section 3.2 (“Global Concept Graph Construction”), we build graphs based on the co-occurrence of topics and knowledge concepts (KCs) extracted from reference documents. Specifically, we construct a topic graph, a topic–KC graph, and a KC graph.
> > 2. As noted in Sec.B of Appendix, the graph used in our experiments contains approximately 32,000 topic nodes and 205,000 KC nodes, retaining only those appearing more than once. The topic–KC graph has ~32,000 edges, ensuring each topic connects to at least one KC. The KC graph includes roughly 22 million edges and has a low density ($1.1 \times 10^{-3}$), indicating a sparse structure.
> > 3. In terms of degree distribution, most topics are linked to 2–5 KCs, with a long tail extending to hundreds for common topics. For KCs, the average degree is around 215. A small number of "hub" KCs have very high degrees, while the majority of KCs have relatively low connectivity. Both degree distributions exhibit heavy-tailed behavior, reflecting the natural co-occurrence patterns in large-scale text corpora.
> > 4. Regarding connectedness, over 98.7% of KC nodes belong to a single giant connected component, while the remaining nodes form only small isolated trees (with no more than four nodes). This high level of connectedness ensures that random walk or depth-first sampling rarely stalls or becomes trapped in disconnected subgraphs.
> > Regarding robustness to graph quality, we adopt several strategies to ensure stable sampling.
> > 5. Beyond the strong connectedness of the graph, we apply frequency-based pruning to remove noisy low-frequency nodes and edges (as mentioned above), making the local graph structure more stable.
> > Additionally, in our weighted random walk, we incorporate a small $\epsilon$-probability (e.g., $10^{-5}$) jump to unvisited nodes (mentioned in Line 258-261). This design helps prevent the sampling process from getting stuck and maintains broad graph coverage.
> >
> > Thanks again for your questions. We will incorporate these analyses and explanations more explicitly in the revised version.
> >
> > -----
> > ### Weakness 6:
> >
> > Thank you for the suggestion. As noted in Table 2 and Lines 320–325 of the paper, we have conducted experiments using a smaller random subset of 3.2M synthetic samples to train our model. As shown in the results, even with only 3.2M examples, our method still outperforms MAmmoTH2 (10M) and JiuZhang3.0 (6M), which are trained on larger datasets. This demonstrates the effectiveness of our approach.

---

> > > ### Author Response · Authors · 2025-06-06
> > >
> > > Dear Reviewer xJ12,
> > >
> > > Thank you very much for your time and thoughtful feedback.
> > >
> > > We look forward to hearing your comments on our response. If there are any remaining questions or concerns, we would be happy to address them further.
> > >
> > > Thank you again for your time and valuable feedback!
> > >
> > > Sincerely,
> > > The Authors

---

> > > > ### Comment · Reviewer_xJ12 · 2025-06-09
> > > > **Official comment by reveiwer**
> > > >
> > > > I wanted to thank the authors for their comprehensive response. I have increased my rating to 6.

---

> > > > > ### Author Response · Authors · 2025-06-10
> > > > > **Response**
> > > > >
> > > > > Dear Reviewer xJ12
> > > > >
> > > > > Thanks for your recognition for our work! We are very glad that our response has addressed your concern. We sincerely appreciate your invaluable feedback throughout the review process and will incorporate your suggestions as we revise the paper.
> > > > >
> > > > > The Authors.

---

### Decision · Program_Chairs · 2025-07-08

**Decision:**

Accept

**Comment:**

This work studies scaling laws for synthetic data for LLMs with a focus on mathematics. The authors propose the SynthLLM as a new scalable framework to transform pre-training data to high-quality and diverse synthetic data better than existing data augmentation methods. There isn't much existing work on scaling laws for synthetic data so this can be a significant contribution to the understanding of synthetic data. Though the work focuses on math, the authors should incorporate the preliminary results on coding and ablations with answer filtering to the final paper as well as more details on the decontamination algorithm used.

Pros:
- Important topic as high-quality data is hard and expensive to obtain.
- Proposes a new synthetic data scaling law to better understand synthetic data.
- Good results on a broad range of math benchmarks.

Cons:
- Less thorough evaluation on prompting.
- Limited to math domain with preliminary results on coding.
- Limited ablations on the effect of wrong answers (e.g. the authors could have ablated with intentionally wrong answers).